# Protease-controlled secretion and display of intercellular signals

Alexander E. Vlahos[1], Jeewoo Kang[2], Carlos A. Aldrete [1], Ronghui Zhu [3], Lucy S. Chong[3], Michael B. Elowitz [3] & Xiaojing J. Gao [1,2✉]

To program intercellular communication for biomedicine, it is crucial to regulate the secretion and surface display of signaling proteins. If such regulations are at the protein level, there are additional advantages, including compact delivery and direct interactions with endogenous signaling pathways. Here we create a modular, generalizable design called Retained Endoplasmic Cleavable Secretion (RELEASE), with engineered proteins retained in the endoplasmic reticulum and displayed/secreted in response to specific proteases. The design allows functional regulation of multiple synthetic and natural proteins by synthetic protease circuits to realize diverse signal processing capabilities, including logic operation and threshold tuning. By linking RELEASE to additional sensing and processing circuits, we can achieve elevated protein secretion in response to "undruggable" oncogene KRAS mutants. RELEASE should enable the local, programmable delivery of intercellular cues for a broad variety of fields such as neurobiology, cancer immunotherapy and cell transplantation.

[1] Department of Chemical Engineering, Stanford University, Stanford, CA 94305, USA. [2] Neurosciences Interdepartmental Program, Stanford University, Stanford, CA 94305, USA. [3] Howard Hughes Medical Institute, Division of Biology and Biological Engineering, California Institute of Technology, Pasadena, CA 91125, USA. ✉email: xjgao@stanford.edu

Synthetic biology aspires to create biomolecular circuits that can sense the state of cells, process the information, and then deliver therapeutic outputs accordingly[1,2]. This vision has been enhanced by the creation of protein-based circuits by others[3–6] and ourselves[7]. Protein-based circuits have advantages such as fast operation, compact delivery, and robust, context-independent performance compared to traditional transcriptional circuits[6,7]. However, these protein circuits have operated in the cytosol, and there remains an urgent need for a design that enables protein-level control of intercellular communication, often required at the "respond" step in "sense-process-respond".

Cell-cell communication is widespread[8–10] and essential for diverse biological processes, such as the generation of immunological responses[11,12], cell differentiation and tissue development[13–15], the maintenance of physiological homeostasis[16], and cancer development[17,18]. Intercellular communication is typically implemented by secreted molecules, including hormones and cytokines. To take cancer immunotherapy as an example, there an ideal application would be introducing a protein circuit that sense the cancerous state of a cell, secrete immunostimulatory signals with temporal and quantitative precision to mobilize the immune system while lysing the cell, and therefore turn these cells into vaccines against other similarly cancerous cells. This would not only avoid the toxic effects associated with the systemic delivery of immuno-modulating proteins[19,20], but also match the complex, dynamic immune process we are trying to control[21]. In contrast, of the current local delivery methods[22], neither nanoparticle[23], or biomaterial-based[24] delivery platforms can fulfill the aforementioned functions that circuits can deliver.

Given the importance of intercellular communication, we sought to interface protein circuits with the secretion and display of protein signals. Specifically, because protease activity has emerged as a "common currency" of protein circuits that responds to synthetic and endogenous inputs[6,7,25–28], it will be ideal to directly control protein secretion using proteases. To design a modular protease-regulated protein secretion system, we adapted aspects of the natural secretion process. Secreted proteins are typically transported into the Endoplasmic Reticulum (ER), processed in the Golgi apparatus, and finally secreted at the plasma membrane. Some proteins contain signaling motifs (e.g., KDEL for soluble proteins[29] and the cytosol-facing dilysine (-KKXX) or -RXR motifs for membrane proteins[30–34]) recognized in the early Golgi apparatus, causing the protein to be retrieved, transported retrogradely, and retained in the ER[32,35]. Other ER-resident proteins, such as cytochrome p450 are retained in the ER via their signal-anchor transfer sequence[36,37]. These retention motifs function in their endogenous contexts as well as when fused to normally secreted proteins[30,33]. To place ER retention under protease control, we engineered the modular Retained Endoplasmic Cleavable Secretion (RELEASE) platform, compatible with both protein secretion and the surface display of membrane proteins. We validated and optimized the core mechanism of RELEASE, created input-processing capabilities, and then used RELEASE to control physiological outputs. Finally, we combined RELEASE with sensing and processing components to respond to internal cell states and external signals via engineered receptors. This study demonstrates a protein-level control module to directly regulate protein secretion that is compatible with pre-existing protein components to program therapeutic circuits for cancer immunotherapy and transplantation in the future.

## Results

### Engineering RELEASE for protein secretion and expression.

RELEASE contains 4 components: an ER-facing linker containing a furin endoprotease cut site, a transmembrane anchor domain[38], a cytosolic linker containing a protease cleavage site, and a cytoplasmic ER retention motif (Fig. 1a, b). On the cytosolic face, the retention motif ensures that the tagged protein is actively transported back to ER[39,40], a process only aborted after the motif is removed by a proteases such as tobacco etch virus protease (TEVP)[6,7]. On the luminal face, soluble proteins are initially tethered to the membrane through the linker and thus coupled to the cytosolic ER retention signal[7]. After the first cytosolic cleavage event, the membrane-tethered protein is transported into the trans-Golgi apparatus, processed into its soluble form (furin is absent in cis-Golgi or ER)[41] (Fig. 1a), and finally secreted.

First, to validate the effectiveness of the retention motif, we fused it to secreted embryonic alkaline phosphatase (SEAP)[42], and used a dilysine-lacking motif (-AAXX) as the negative control (not retained). We transiently transfected human embryonic kidney (HEK) 293 cells using DNA plasmids encoding the constructs and 48 h later removed the supernatant to measure SEAP activity following incubation with a colorimetric substrate. Using RELEASE, SEAP is minimally present in the supernatant and comparable to control cells that were not transfected with SEAP (Fig. 1c). Switching the dilysine retention motif (-KKXX) with -AAXX significantly increases SEAP secretion (Fig. 1c). We next placed the dilysine motif under the control of TEVP and showed that the co-expression of TEVP significantly increases SEAP secretion (Fig. 1d—left panel). By switching the cytosolic protease cut sites, we validated RELEASE against other orthogonal proteases such as the hepatitis C virus protease (HCVP) (Fig. 1d—right panel) and the tobacco mottling vein virus protease (TVMVP) (Supplementary Fig. 1a, b). Furthermore, the design is compatible with alternative ER-retention motifs, as we validated constructs using the N-terminal signal anchor sequence from p450[36,37] (Supplementary Fig. 2a, b), which confers retention by directly inserting into the ER membrane[36,37] rather than retention through retrograde transport[31,32].

In anticipation of tuning RELEASE for different applications, we next explored how its performance is affected by two design decisions. First, as an alternative to the tri-transmembrane domain[38], we created a single transmembrane variant, and found it more sensitive to TEVP compared to the tri-transmembrane construct (Supplementary Fig. 3a). Similarly, the input sensitivity of HCVP-inducible RELEASE is also modulated by the choice of the transmembrane domain (Fig. 1e). Furthermore, by using a protein linker containing the native residues that flank the HCVP cut site[38,43], we made more sensitive HCVP-inducible RELEASE constructs (Fig. 1e—red and green lines) than the original versions that use synthetic flanking sequences. A complete list of the cleavage efficiencies for the RELEASE variants are in Supplementary Table 1. We took advantage of this tunability to reduce RELEASE response to the input-independent activity of a membrane-localized split protease[44] (Supplementary Fig. 3b) and therefore improve output dynamic range (Supplementary Fig. 3c, d).

Dual-input systems for fine tuning signaling pathways are an important tool in synthetic biology[45] and we hypothesized that RELEASE could be used with traditional inducible gene expression systems to achieve this. Since RELEASE controls protein secretion post-translationally, we used SEAP-RELEASE and rapalog-inducible split TEVP constructs, under the control of a doxycycline-inducible promoter, to measure SEAP secretion with different combinations of inducers (Fig. 1f). Indeed, SEAP secretion was dependent on the presence of both inducers, and the biggest increase in SEAP secretion was observed when both rapalog and DOX was present relative to when they were both absent (Fig. 1f—76.9-fold increase).

We also compared the dynamics of SEAP secretion using RELEASE to traditional inducible gene expression systems

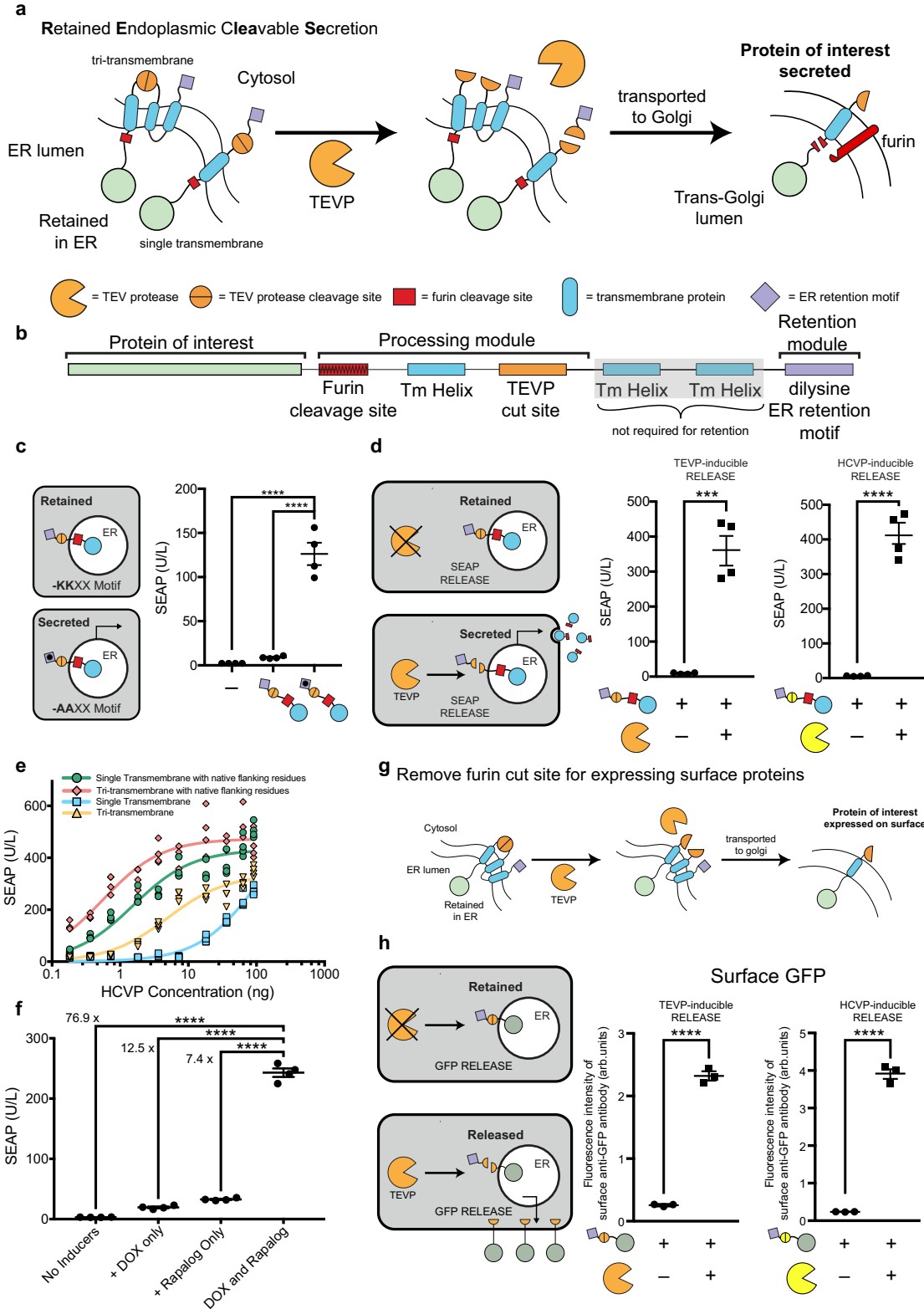

**a** Retained Endoplasmic Cleavable Secretion

= TEV protease    = TEV protease cleavage site    = furin cleavage site    = transmembrane protein    = ER retention motif

**b**

Protein of interest | Processing module | Retention module

Furin cleavage site | Tm Helix | TEVP cut site | Tm Helix | Tm Helix | dilysine ER retention motif

not required for retention

**g** Remove furin cut site for expressing surface proteins

**h** Surface GFP

(Supplementary Fig. 4a). Following transfection using a DOX-inducible SEAP, there was a significant difference in protein secretion observed 3 h after stimulation with DOX, compared to the uninduced controls (Supplementary Fig. 4b). In comparison, SEAP-RELEASE co-expressed with a rapalog-inducible split-TEVP showed a significant difference in protein secretion, 3 h post-induction with rapalog (Supplementary Fig. 4c), which could be

improved to 2 hours by overexpressing furin (Supplementary Fig. 4d). This suggested that furin cleavage could be one of the factors limiting the speed of protein secretion using RELEASE.

In addition to controlling protein secretion, cells can communicate by changing the display of proteins on their surface[12,13]. By removing the furin cut site in RELEASE, we hypothesized it could control the surface display of proteins (Fig. 1g). To validate this

**Fig. 1 Design of Retained Endoplasmic Retained Secretion (RELEASE). a** Proteins of interest are fused to RELEASE and retained in the ER via the dilysine ER retention domain (purple diamond). Upon activation or expression of a protease such as TEVP (orange partial circle), the ER retention domain is removed (middle panel) and the protein of interest is transported through the constitutive secretory pathway. When reaching the Trans-Golgi Apparatus (right panel), the native furin endoprotease cleaves the linker region allowing the membrane-bound protein to be secreted. **b** RELEASE is a modular platform and can be modified to respond to different proteases and regulate different proteins of interest. **c** The C-terminal dilysine motif of RELEASE is required for SEAP retention and using a construct where the lysine residues were modified to alanines (KKXX-COOH → AAXX-COOH) increased SEAP secretion. There was no significant difference in signal between RELEASE and control cells without SEAP. **d** Co-expression of proteases such as TEVP (orange partial circle), or HCVP (yellow partial circle) with the respective RELEASE constructs increased SEAP secretion. **e** The cleavage efficiencies of HCVP RELEASE constructs were also affected by transmembrane selection and was improved by modifying the residues flanking the HCVP cut site with native linker proteins. Based on the steady-state solution of a kinetic model for proteolytic cleavage, we determined that the relation between RELEASE output and the amount of protease plasmids fits the Michaelis–Menten equation[7]. We therefore fit the titration curves using Michaelis–Menten equations and used $K_m$ to represent the apparent cleavage efficiency of each design by its corresponding protease. A complete list of the calculated cleavage efficiencies for the different RELEASE constructs can be found in Supplementary Table 1. **f** Dual-input control of protein expression was achieved using traditional DOX-inducible systems and RELEASE. Using a DOX-inducible SEAP-RELEASE with a Rapalog-inducible split TEVP, we observed fine control of SEAP secretion over a large dynamic range. **g** By removing the furin cut site, RELEASE was amenable to control the surface display of proteins. **h** Increased surface display of membrane-bound GFP fused to RELEASE in response to TEVP (left panel) or HCVP (right panel). Each dot represents a biological replicate. Mean values were calculated from four (**c–f**) or three replicates (**h**). The error bars represent ±SEM. The results are representative of at least two independent experiments; significance was tested using an unpaired two-tailed Student's $t$-test between the two indicated conditions for each experiment. For experiments with multiple conditions, a one-way ANOVA with a Tukey's post-hoc comparison test was used to assess significance. ***$p < 0.001$, ****$p < 0.0001$.

strategy, membrane-bound green fluorescent protein (GFP) fused to RELEASE and a mCherry co-transfection marker were transfected into HEK293 cells, and the cell surface was stained using an anti-GFP antibody. To quantify changes in the surface display of GFP, the mCherry co-transfection marker was used to select for highly transfected cells, which show the largest separation of reporter fluorescence from cellular autofluorescence (Supplementary Fig. 5). GFP-RELEASE constructs significantly increased surface display of GFP in response to the cognate proteases (Fig. 1h). Taken together, these results show that RELEASE is a suitable approach to control the secretion and surface display of proteins in response to protease activity (Fig. 1d, h).

**RELEASE is compatible with circuit-level functions.** After validating the RELEASE design, our next goal was to ensure that its activation could be programmed using protease-based circuits, either pre-existing[6,7] or novel. For RELEASE to operate properly in circuits with multiple proteases, first it is important to validate the orthogonal control of RELEASE by the selected protease[7]. Indeed, cells simultaneously transfected with two RELEASE constructs (Fig. 2a) were orthogonal and only secreted the respective reporter protein in response to the cognate protease (Fig. 2b). In addition, we observed a slight reduction in SEAP and GFP secretion when both proteases were expressed, relative to when only a single protease was expressed, potentially due to protein overexpression draining finite cellular resources[46]. This result demonstrates that two proteases can be used to independently regulate secretion of distinct target proteins in the same cell.

In addition to the parallel regulation of multiple outputs, another useful capability is the integration of multiple inputs. Logic operation is crucial for integrating multiple signals, previously implemented for protease circuits using degrons[7], or coiled-coiled (CC) dimerization domains[6]. RELEASE enables the compact implementation of Boolean logic directly at the retention level. To implement OR, two protease cut sites were inserted in tandem into the cytosolic linker so that the retention motif is removed by either protease (Fig. 2c). To implement AND, a RELEASE complex was created containing the N-terminal p450 signal anchor sequence and the C-terminal dilysine motif, each alone conferring sufficient ER retention (Fig. 2d). For SEAP to be secreted, both motifs must be removed (Fig. 2d). We

attributed the reduced secretion in the AND gate construct due to the use of the N-terminal signal anchor sequence (Supplementary Fig. 2b). Both gates function as expected (Fig. 2c, d). We also implemented an alternative approach for AND (Supplementary Fig. 6)[47,48].

Other than processing signals on its own, can RELEASE be coupled to other protease circuits? We used protease-activated protease as an example of such circuits[6]. We used CC domains to associate split protease halves with complementary CC domains and catalytically inactive halves (Fig. 2e), "caging" them by preventing the active halves from associating with each other. Cut sites were incorporated between the CC domains and between the CC domain the inactive half, allowing the input protease to remove the inhibitory domains. Following removal of the autoinhibitory portion, the complementary CC domains of the functional split protease halves would then associate and reconstitute protease activity (Fig. 2e). Using this approach, we created a two-protease cascade, in which TEVP activates TVMVP, which in turn cleaves the TVMVP-inducible RELEASE. This circuit increased SEAP secretion in response to TEVP, while maintaining strong retention in the absence of TEVP (Fig. 2f). This highlights the modularity of the RELEASE design and the ability to engineer additional functionality into it.

**RELEASE controls biologically relevant proteins.** Many cytokines are pleiotropic and their systemic administration would cause serious adverse effects, so controlling their local expression with RELEASE would be advantageous for tumor immunotherapy[19]. We selected interleukin 12 p70 (referred to as IL-12), because it is a immunomodulatory cytokine important for T-cell activation and proliferation[49,50]. IL-12 is composed of two obligatory subunits (p35 and p40)[51], so we fused the two subunits with a flexible linker[19,52] and then with RELEASE (Fig. 3a). As expected, TVMVP significantly increases IL-12 secretion (Fig. 3b) with therapeutically relevant concentrations for cancer immunotherapy[19].

As for controlling membrane proteins, we chose the Kir2.1 potassium channel as an example of (Fig. 3c), because it is a powerful tool in neurobiology[53,54] and a well-characterized model membrane protein. A protease-controlled Kir2.1 would enable the conditional silencing of neurons based on their intracellular states or extracellular cues, e.g., therapeutic silencing of the most active neurons during a seizure without the side effects of conventional methods that exert indiscriminate silencing. Unlike

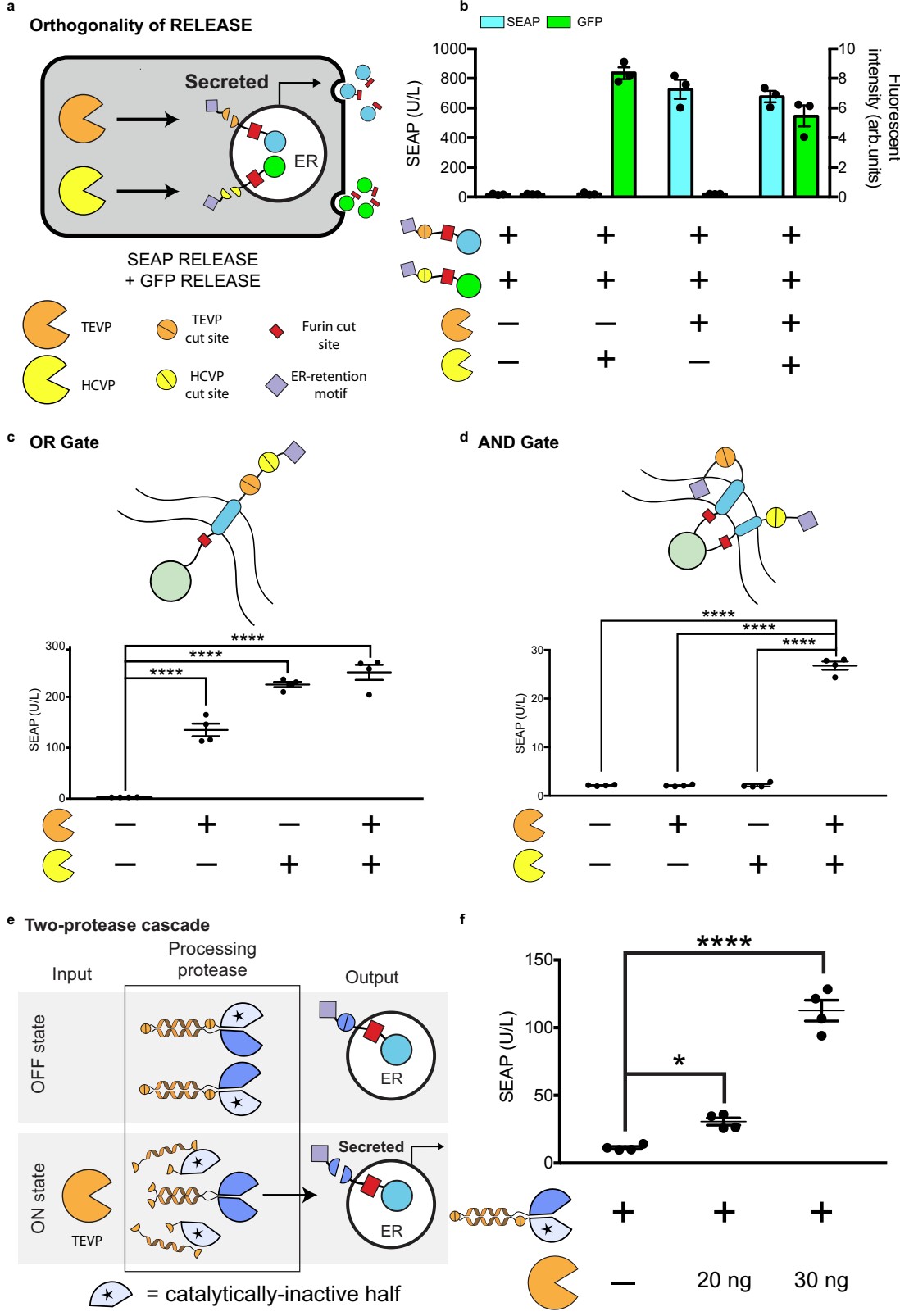

secreted proteins, Kir2.1 has a cytoplasmic-facing ER export motif (FXYENEV)[55] that directs its transport through the secretory pathway, posing a potentially unique challenge in the retention ability of RELEASE and will serve as a test case for its future adaptation to other membrane proteins. To measure the surface display of Kir2.1, a hemagglutinin (HA) epitope was incorporated into its extracellular loop[34]. In addition, GFP or mCherry was fused to the N-terminus of Kir2.1, to select cells that were highly expressing Kir2.1 for flow cytometry analysis (Supplementary Fig. 7a). Initial experiments fusing Kir2.1 with the standard RELEASE construct resulted in leaky display of Kir2.1 in the absence of TEVP (Supplementary Fig. 7b). We

**Fig. 2 RELEASE in circuits. a** Orthogonal operation of RELEASE constructs. **b** HEK293 cells were co-transfected with SEAP fused to RELEASE (responsive to TEVP) and GFP fused to RELEASE (responsive to HCVP). SEAP and GFP levels increase in the supernatant when the cognate protease was expressed. **c** Tandem insertion of two protease cut sites (top panel) created a RELEASE construct that implemented OR gate logic. If either of the respective proteases were expressed, the dilysine ER retention motif would be removed, and SEAP would be secreted. **d** Implementation of AND logic by adding the N-terminal p450 signal anchor sequence as a second ER retention domain, so that both proteases would have to be present to remove both retention domains and allow SEAP to be secreted. **e** A two-protease cascade was created where TEVP was required to activate TVMVP (processing protease), which subsequently cleaved SEAP RELEASE. SEAP secretion increased when TEVP was expressed (right panel). Each dot represents an individual biological replicate. Mean values were calculated from three (**b**) or four replicates (**c–e**). Error bars represent ±SEM. The results are representative of at least two independent experiments; significance was tested by one-way ANOVA with a Tukey's post-hoc comparison test among the multiple conditions. *$p < 0.05$, ****$p < 0.0001$.

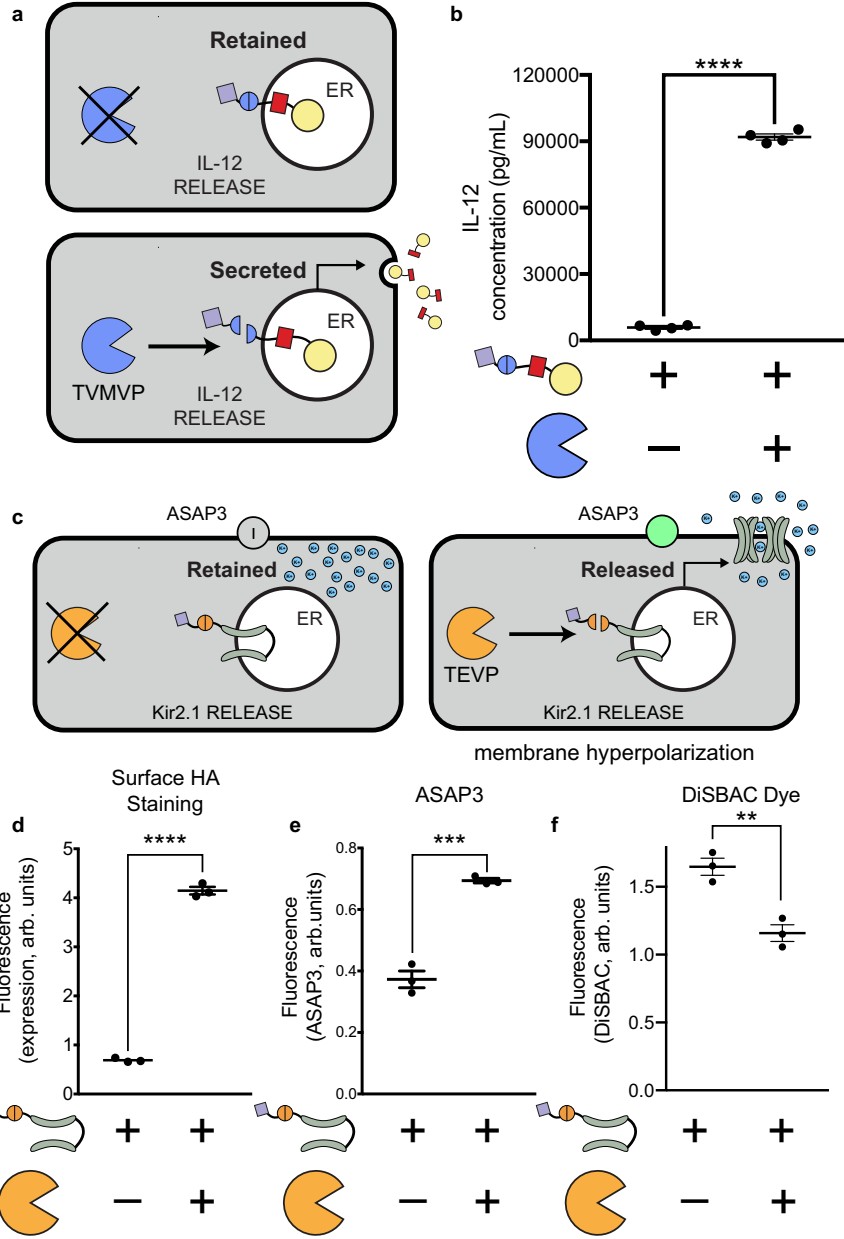

**Fig. 3 Controlling bioactive proteins using RELEASE. a** The cytokine IL-12 was fused to RELEASE and placed under the control of TVMVP. **b** TVMVP significantly increase IL-12 secretion. **c** The inwardly rectifying potassium channel Kir2.1 was fused to RELEASE. In addition, the genetically encoded voltage indicator ASAP3 was co-transfected. **d** Co-expression of Kir2.1-RELEASE with TEVP resulted in a significant increase in the amount of Kir2.1 expressed on the surface, which was quantified using surface staining for HA and flow cytometry. The surface display of functional Kir2.1 in response to TVMVP was shown to cause hyperpolarization of transfected cells. This was validated by measuring change in the fluorescence intensity of the genetic reporter **e** ASAP3, or the chemical dye, **f** DiSBAC2(3). Each dot represents an individual biological replicate. Mean values were calculated from four (**b**) or three replicates (**d–f**). Error bars represent ±SEM. The results are representative of at least two independent experiments. Significance was tested using an unpaired two-tailed Student's $t$-test between the two indicated conditions for each experiment. **$p < 0.01$, ***$p < 0.001$, ****$p < 0.0001$.

reasoned that it is because Kir2.1 has a long cytosolic tail, and that the dilysine motif is the most effective when positioned closely to the ER membrane[30,34]. In contrast, another ER retention motif, RXR, is most effective when positioned distally from the membrane[34]. Indeed, a RELEASE construct using the RXR motif at the C-terminus improved retention (Supplementary Fig. 7), and successfully controlled its surface display using TEVP (Fig. 3d).

Kir2.1 functions as a homo-tetramer[56], provoking the question of whether the RELEASE system could interfere with tetramerization and consequently channel function (Fig. 3c). Surface display of functional Kir2.1 leads to efflux of potassium ions and hyperpolarization[56], providing a metric we can use to assess the its functionality. We used two reporters to measure changes in membrane potential: ASAP3[57] and DiSBAC$_2$(3)[58]. ASAP3 is a genetically encoded voltage indicator[57] that increases fluorescence as cells become hyperpolarized, while DiSBAC$_2$(3) is a chemical dye that decreases diffusion into hyperpolarized cells, and therefore fluorescence intensity[59]. When Kir2.1 RELEASE was co-expressed with ASAP3, we observed a significant increase in fluorescence intensity in response to TEVP (Fig. 3e), suggesting Kir2.1 was functional. The chemical dye DiSBAC$_2$(3) showed similar results (Fig. 3f), and the observed change in median fluorescent intensity that cells were hyperpolarized[58], further corroborating that RELEASE-regulated Kir2.1 maintains its functionality.

**RELEASE responds to oncogenic inputs.** One of the most compelling cases for protein circuits is therapy against recalcitrant cancers. The RAS family of proteins (HRAS, KRAS, and NRAS) provide a remarkable example[60,61]. The activating RAS mutations (e.g. G12V, G12C, G12D…) have been implicated in a multitude of hard-to-treat cancers such as pancreatic ductal adenocarcinoma[60–62] and non-small lung cancer[63]. The pharmacological targeting of RAS has been challenging[64–66]. We envision a "circuit as medicine" alternative, where an intracellularly introduced circuit interrogates the cancerous state of a cell, and conditionally lyses RAS-mutant cells, while programming cytokine secretion to activate a broader local immune response.

As a first step towards that vision, we hypothesized that we could exploit protein interaction during RAS signaling to activate RELEASE. RAS resides in the cell membrane[67,68], and activated RAS recruits to the membrane effector proteins such as Raf[68–70]. To sense active RAS, we fused the N- and C-terminal halves of split TEVP to the RAS-binding domain (RBD) of Raf (Fig. 4a). The increased local concentration of the RBD-split TEVP sensor in response to activated RAS, due to their transition from the 3D cytosol to the more restrictive 2D membrane, was expected to facilitate the association of the protease halves through their residual mutual affinity[67].

Building on our previous constructs sensing the RAS pathway[7], we performed initial experiments using the active mutant HRAS-G12V and the RBD-split TEVP sensor, and observed comparable SEAP secretion with wildtype cells (endogenous HRAS activity) when regulated by TEVP-responsive RELEASE (Supplementary Fig. 8). Since HRAS-G12V reconstitutes RBD-split TEVP at the cell membrane, and cleavage of RELEASE occurs at the ER, we hypothesized that additional protease components would be required to propagate the signal from the cell membrane to the ER (Fig. 4a). Using the caged TVMVP intermediate protease (Figs. 2d, 4b—topology 1) did not improve SEAP secretion in response to HRAS-G12V, so we further hypothesized that spatial localization of the intermediate protease might be required to increase signal transduction. We first tried to increase the cleavage of the intermediate protease by bringing it closer to the

TEVP input, fusing the C-terminal membrane transfer CAAX motif[71] (Supplementary Fig. 9a—left panel) to one half of the caged split TVMVP (Fig. 4b—topology 2), but this did not improve SEAP secretion (Fig. 4c). We then also increased the possibility for the reconstituted intermediate protease to activate RELEASE, by fusing the uncaged other half of TVMVP with the signal anchor sequence of cytochrome p450 and therefore targeting it to the ER membrane (Fig. 4b—topology 3). This resulted in the greatest SEAP secretion in response to HRAS-G12V (Fig. 4c). After titrating down the ER-bound uncaged half of TVMVP, we reduced background and improved dynamic range (Supplementary Fig. 9b).

We then generalized the design to KRAS, the most frequently mutated RAS in cancer[72]. We validated that our circuit responds very similarly to KRAS-G12V and HRAS-G12V (Supplementary Fig. 9c), probably because RAS isoforms share up to 90% homology in the region where RBD binds[64,73]. As a control, the split TEVP sensor fused to the RBD mutant (R89L), which has a reduced affinity to activated RAS[67,70], did not significantly increase SEAP secretion in response to HRAS-G12V or KRAS-G12V (Fig. 4e).

We reasoned that the choice of cell membrane-localization domains might affect baseline, because post-translational modification of CAAX initially inserts the protein at the ER membrane[74], which could facilitate TVMVP reconstitution in the absence of TEVP inputs. To further reduce the background of the RAS sensor, we additionally tested the N-terminal membrane anchoring portion of the SH4 domain of Lyn and Fyn tyrosine kinases[75], the cell membrane-targeting of which bypasses ER[75]. The Lyn and Fyn motifs reduced background SEAP secretion relative to the CAAX motif (Supplementary Fig. 9d) and enabled increased SEAP secretion without significantly increasing the background (Supplementary Fig. 9e).

The complete circuit is summarized in Fig. 4d. We observed that the circuit was responsive to the oncogenic state of KRAS, since cells secreted significantly more SEAP when overexpressing active mutants of KRAS (Fig. 4e—blue bar, Supplementary Fig. 9g) compared to overexpressing wildtype KRAS (Fig. 4e—green bar) and endogenous wildtype KRAS (Fig. 4e—red bar). The oncogenic state of KRAS also resulted in a much smaller increase in SEAP secretion when using the RBD-split TEVP R89L mutant (Fig. 4e).

**Plug-and-play capabilities of RELEASE.** In addition to building towards RAS detection, our RAS-centric engineering efforts also established a plug-and-play protein circuit framework. RELEASE, in conjunction with CHOMP and other protease components, enables the detection of any input that can be converted to dimerization or proteolysis. This signal can then be processed by RELEASE itself or other protease circuits to control the display or secretion of proteins (Fig. 5a).

As a proof of principle, we used the well-established MESA receptor (membrane-localized split TEVP reconstituted by rapalog)[25,26,76] as an input to activate RELEASE via the intermediate protease circuit optimized above (Fig. 4c). Switching the input components to the rapalog MESA receptor, we increased SEAP secretion in response to rapalog (Fig. 5b). We also used RELEASE to control the secretion of IL-12 in response to mutant KRAS (Fig. 5c) or rapalog (Fig. 5d), and to control the surface display of Kir2.1 by rapalog (Fig. 5e).

The processing protease circuit is also modular. Specific applications of RELEASE may require a greater dynamic range or more complex dynamic secretion patterns that can be achieved by incorporating additional orthogonal proteases[6,7]. For example, to improve the dynamic range of the RAS-sensing circuit, we

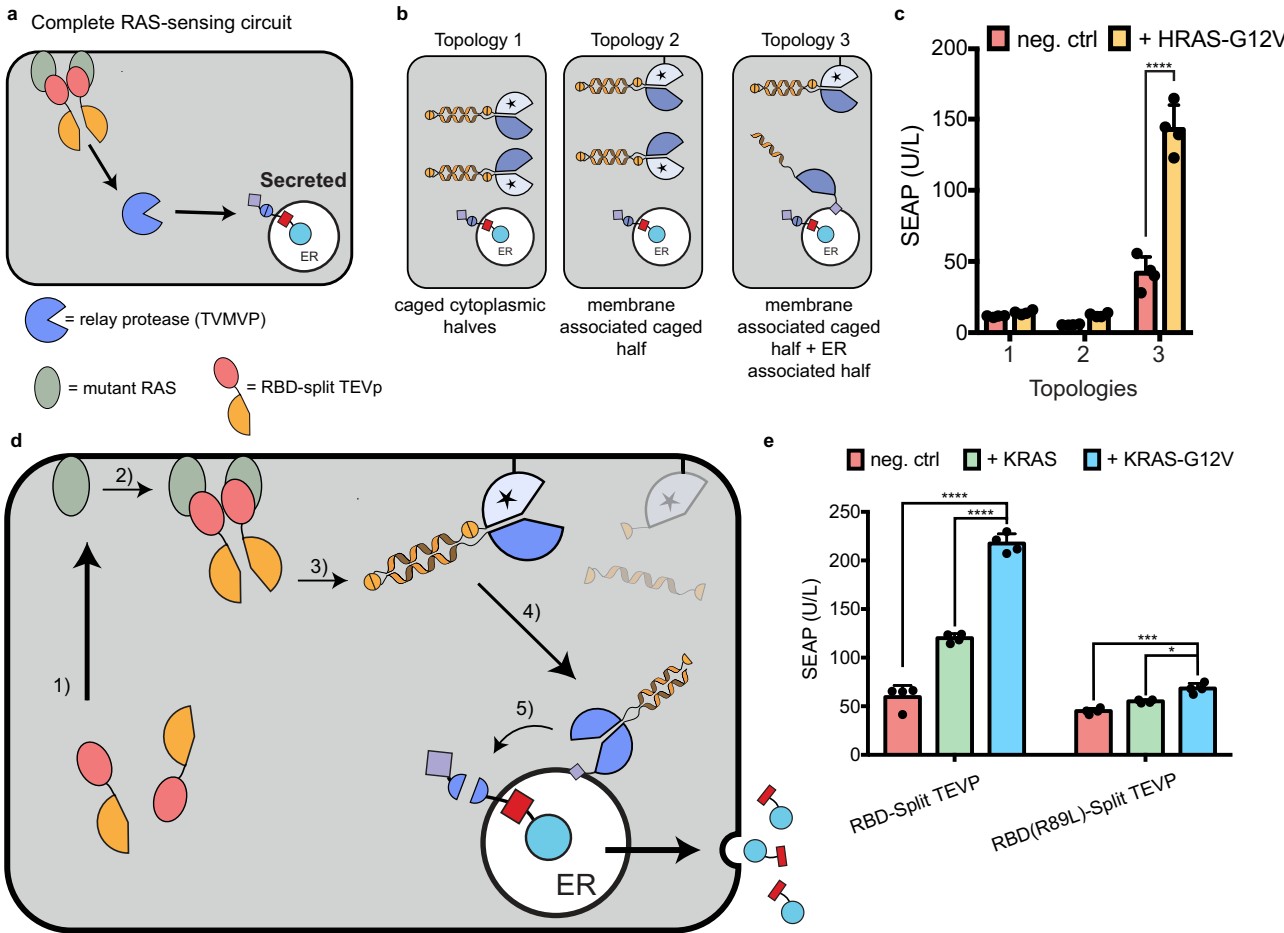

**Fig. 4 RAS-sensing circuit and protease replaying pathways to activate RELEASE. a** To sense active RAS, split TEVP was fused to the RBD domain of c-RAF. RBD-split TEVP binds to active RAS at the membrane surface of the cell where the two protease halves reassociated and reconstituted protease activity. Protease activation is propagated through an intermediate protease to relay the information from the cell membrane to the ER. **b** Using protein localization motifs, three different topologies of intermediate protease components were created. Topology 1 uses two caged intermediate TVMVP protease halves in the cytosol. Topology 2 uses the same caged intermediate TVMVP, but with one half of the active protease localized to the membrane. Finally, Topology 3 has one half of the intermediate protease associated with the membrane, and the other half uncaged and present at the ER membrane via the p450 signal anchor sequence. The CC domain present on the uncaged TVMVP half (that was associated with the membrane) drives association with the complementary TVMVP half at the ER. **c** There was a significant difference in the amount of SEAP secreted when using intermediate protease topology 3, with and without mutant HRAS-G12V, compared to topologies 1, and 2. **d** Schematic of the signal processing of the complete KRAS-sensing circuit. The complete RAS-sensing circuit was activated by RBD-split TEVP interacting with active KRAS-G12V (1). The reconstituted TEV (2) then uncaged the membrane associated split TVMVP, releasing it from the membrane (3). The uncaged TVMVP contains a CC domain, which drives its association with the complementary CC domain present on the other split TVMVP half anchored to ER membrane (4). Finally, the reconstituted TVMVP cleaves the ER retention motif of RELEASE to secrete SEAP (5). **e** Using the complete RAS-sensing circuit, we observed a significant increase in SEAP secretion when expressing an active mutant variant KRAS-G12V relative to baseline levels (neg. ctrl), or wildtype KRAS. A small, but statically significant increase was also observed when using the RBD-Split TEVP containing the R89L mutation that reduced the association with active KRAS. Each dot represents an individual biological replicate. Mean values were calculated from four replicates (**c**, **e**). The error bars represent ±SEM. The results are and representative of at least two independent experiments. Significance was tested using an unpaired two-tailed Student's *t*-test between the two indicated conditions for each experiment. For experiments with multiple conditions, a one-way ANOVA with a Tukey's post-hoc comparison test was used to assess significance. **\*\****p* < 0.01, \*\*\**p* < 0.001, \*\*\*\**p* < 0.0001.

incorporated a previously established positive feedback loop based on reciprocal inhibition between TVMVP and HCVP to tune the activation threshold of TVMVP[7] (Fig. 5f). When input was low or absent, HCVP would reduce the "baseline" reconstitution of TVMVP by removing the complementary CC domain (Fig. 5f—top panel). However, when there was sufficient input (KRAS-G12V$^+$ cells), the reconstituted TVMVP would override HCVP by removing its activity-enhancing co-peptide (Fig. 5f—bottom panel). By varying the amount of HCVP transfected, we reduced the background activity and increased the dynamic range of the engineered cells containing the complete

RAS circuit (Fig. 5g). These results demonstrate the possibility of tuning RELEASE with additional proteases and eventually creating more complex responses.

## Discussion

Here, we introduced the generalized protease-responsive platform RELEASE to control the secretion and display of proteins (Fig. 1). RELEASE is compatible with protein-level circuit operations (Fig. 2) and enables plug-and-play control of various outputs (Figs. 3 and 5) using a variety of inputs (Figs. 4 and 5). For all

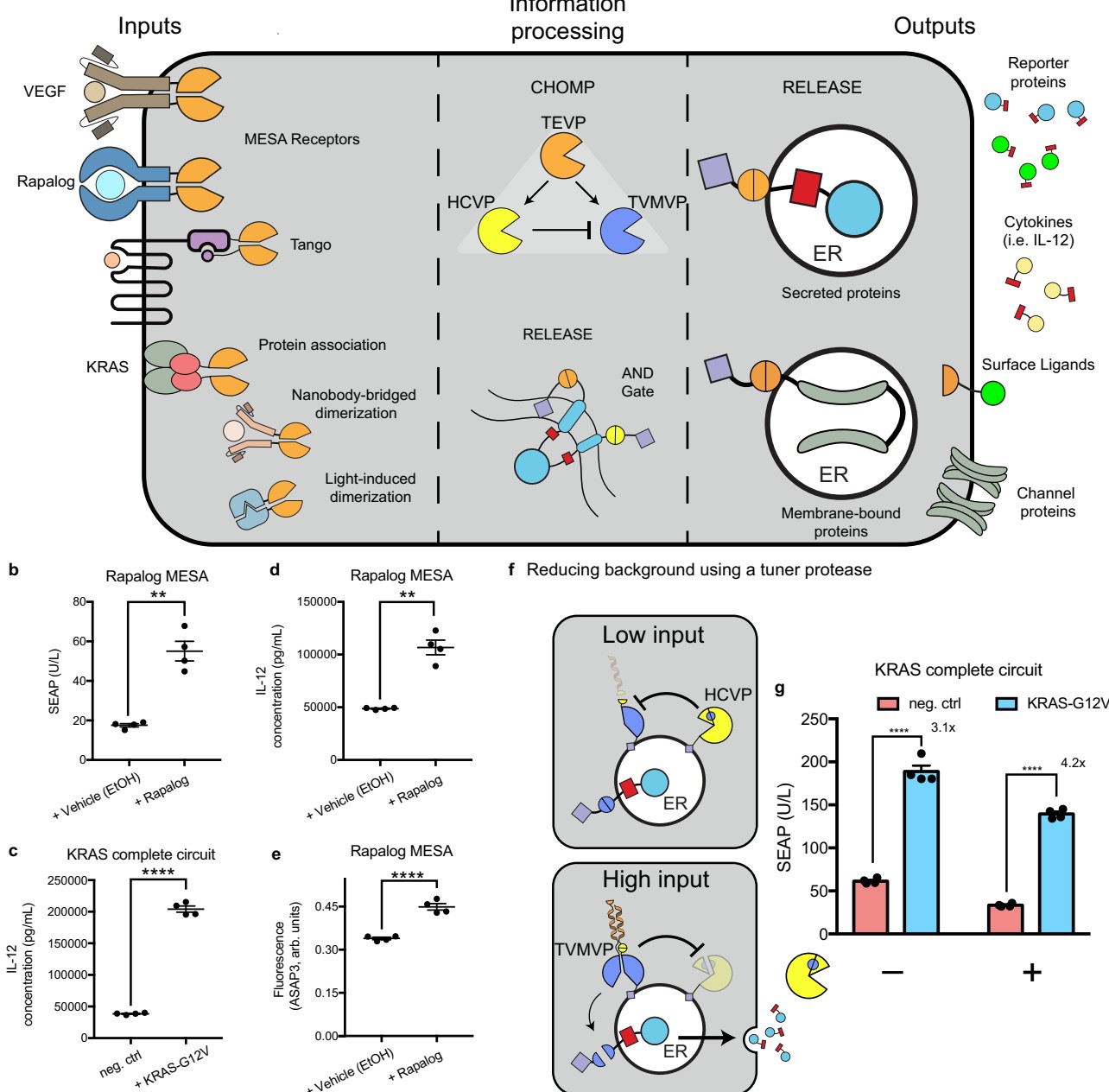

**Fig. 5 Plug-and-play capabilities of RELEASE. a** Any multimerization event, such as ligand-induced receptor dimerization (i.e. MESA receptors, or Tango), protein association, nanobody-bridged dimerization, or light-induced dimerization can be harnessed to reconstitute and activate split proteases. This information can then be processed using CHOMP circuits or even RELEASE itself to produce complex responses. Each component of the engineered circuit can be optimized independently of each other and are not necessarily dependent on the input or output components. To highlight the plug-and-play capabilities of RELEASE, we tested different input and output combinations, while keeping the intermediate CHOMP circuit intact. **b** Using the rapalog MESA receptor as the input, SEAP secretion was controlled. IL-12 secretion was induced by **c** KRAS expression or induction with **d** rapalog. **e** We also observed Kir2.1-mediated hyperpolarization after induction with Rapalog. **f** Schematic of CHOMP circuit containing reciprocal inhibition of TVMVP and HCVP to reduce background activity of RELEASE. When the amount of input is low, the ER-associated split TVMVP protease is repressed by the ER-associated HCVP through removing the complementary CC motif, reducing the association with the other split functional half. When the amount of input is high, fully reconstituted TVMVP will be present at higher levels and repress HCVP by removing the core HCVP from its activity-enhancing co-peptide (small yellow pie space). **g** Addition of the tuner protease increased the dynamic range of the RAS-sensing circuit, by reducing baseline secretion. Each dot represents an individual biological replicate. Mean values were calculated from four biological replicates (**b**–**e**, **g**). Error bars represent ±SEM. The results are representative of at least two independent experiments. Significance was tested using an unpaired two-tailed Student's $t$-test between the two indicated conditions for each experiment. **p < 0.01, ****p < 0.0001.

these examples, we simply switched the input and output (RELEASE) components, while keeping the intermediate protease chassis intact—without any re-optimization. This highlights the modularity of using protease-based sensors, protease circuits, and RELEASE to engineer sense-and-response capabilities. In contrast to traditional small molecule-induced reporter systems, RELEASE enables close-loop control in biomedical contexts by sensing cellular states, processing the information, and actuating suitable outputs accordingly, all at the level of engineered protein circuits.

When adapting RELEASE for new applications, all one needs is a protein-mediated dimerization event that could be harnessed to reconstitute protease activity (Figs. 4a and 5a). We can therefore tap into additional synthetic receptor platforms that rely on ligand-induced dimerization, such the Generalized Extracellular Molecule Sensor (GEMS)[42], or Tango[77]. RELEASE is also compatible with traditional gene expression systems (i.e. DOX) to form dual-input systems that offer stricter dynamic control over protein secretion or surface expression (Fig. 1f). In this work we demonstrate that one can use intermediate proteases to propagate protease signal from the cell membrane to the ER to activate RELEASE (Fig. 4a), suggesting that using alternative motifs may allow for signal propagation from other subcellular locations, such as nucleus or mitochondria, to ER. Because the components of the conventional protein secretion pathway are conserved among different cell types and species, we expect RELEASE to function in these different contexts as well. If the new application requires a faster response, we could improve the cleavage efficiency of the furin cut site of RELEASE, such as using different furin cut sites or using a protein linker with multiple furin cut sites in tandem, as furin was implicated to be a limiting factor in the speed of RELEASE (Supplementary Fig. 4c, d).

RELEASE enables therapeutic modalities. For example, we hypothesize that we can use the KRAS-sensing circuit (Fig. 4c) to selectively express immunostimulatory signals (such as IL-12, surface T-cell engagers, and anti-PD1) to mark cancer cells for T cell-mediated destruction without affecting normal cells[19]. The selectivity of the circuit will be further improved using additional proteases through quantitative thresholding (Fig. 5g) or logic operations. For the latter, many RAS-driven cancers harbor additional mutations in tumor suppressor proteins, such as p53[78]. One could use split proteases fused to nanobodies[4,7] that have preferential binding to mutant p53[78], to activate RELEASE only when both mutant KRAS and mutant p53 are simultaneously present, via AND logic (Fig. 2c). In this work, we validated our therapeutic circuits using HEK293 cells overexpressing active mutant (Fig. 4e). However, future studies must be completed using cancer cells harboring the active KRAS mutants (i.e. PANC 10. 05, PANC 03. 27, or LS180) to demonstrate significant differences compared to endogenous KRAS activity in healthy cells, in vitro and using in vivo xenograft models.

Delivery will be another barrier that will have to be overcome for RELEASE to be useable for cancer immunotherapy. An additional benefit of RELEASE compared to traditional synthetic circuits is that protease circuit components can be encoded within single mRNA transcripts[79] that do not pose the risk of insertional mutagenesis. Furthermore, combinatorial immunomodulators have been found to be more effective than monotherapies for targeting cancer cells[19]. Due to the complexity and broad activity of the immune system, we anticipate that we will need to control the secretion and surface display of multiple immunostimulatory proteins, which is possible due to the modularity of RELEASE (Figs. 2b and 3b).

RELEASE will also expedite other potential therapeutic applications in fields as diverse as neurobiology, developmental biology, immunology, tissue engineering, and transplantation, to name a few. To take a third and last example, in addition to the

cancer immunotherapy and neuronal silencing applications discussed above, RELEASE can be used to create sense-and-respond cells to control immunomodulating cytokines and growth factors important for graft acceptance, such as IL-10[80] and TGF-β[20], which cannot normally be delivered systemically due to their pleiotropic and off-target effects. Co-delivering these engineered cells with therapeutic cells, such as pancreatic islets, may be a suitable approach to create engineered tissue implants that can engraft without the need for systemic immunosuppression. The proposed plug-and-play components for sensing and secreting using RELEASE would allow for the programming of such communications with unprecedented precision.

## Methods

**Plasmid generation**. All plasmids were constructed using general practices. Backbones were linearized via restriction digestion, and inserts were generated using PCR, or purchased from Twist Biosciences. MESA-rapalog receptor source plasmids were a generous gift from Joshua Leonard[26]. The plasmid containing the voltage indicator, ASAP3 was a generous gift from Michael Lin[57]. A complete list of plasmids used for each experiment (Supplementary Data 1) and the respective amounts used for transfections can be found in supplementary data 2. In addition, the DNA sequences of all the plasmids used in this study can be found in the source data, and all new plasmids with annotations will be available on Addgene (https://www.addgene.org/Xiaojing_Gao/).

**Tissue culture**. Flp-In™ T-REx™ Human Embryonic Kidney (HEK) 293 cells were purchased from Thermo Fisher Scientific (catalog# R78007). Cells were cultured in a humidity-controlled incubator under standard culture conditions (37 °C with 5% $CO_2$) in Dulbecco's Modified Eagle Medium (DMEM), supplemented with 10% fetal bovine serum (FBS - Fisher Scientific; catalog# FB12999102), 1 mM sodium pyruvate (EMD Millipore; catalog# TMS-005-C), 1X Pen-Strep (Genesee; catalog# 25-512), 2 mM L-glutamine (Genesee, catalog# 25-509) and 1X MEM non-essential amino acids (Genesee; catalog# 25-536). Cells tested negative for mycoplasma.

To induce expression of transiently transfected plasmids, 100 ng/mL of Doxycycline was added at the time of transfection. All measurements were taken 48 h after transient transfection, unless otherwise stated. Rapalog AP21967 (also known as A/C heterodimerizer, purchased from Takara Biosciences; catalog# 635056) is a synthetic rapamycin analog that can bind with FRB harboring the T2098L mutation, and is designed not to interfere with the native mTOR pathway[81]. All our constructs in this study using the FRB protein contain the T2098L mutation and were induced with 100 nM of rapalog, unless otherwise stated.

**Transient transfections**. HEK 293 T cells were cultured in either 24-well or 96-well tissue culture-treated plates under standard culture conditions. When cells were 70-90% confluent, the cells were transiently transfected with plasmid constructs using the jetOPTIMUS® DNA transfection Reagent (Polyplus transfection, catalog# 117-15), as per manufacturer's instructions.

**Measuring protein secretion**. Secreted Alkaline Phosphatase (SEAP) Assay was performed to measure protein secretion[42]. Briefly, following two days after transient transfection, the supernatant was collected without disrupting the cells and heat inactivated at 70 °C for 45 min. Following heat inactivation, 10–40 μL of the supernatant was mixed with $dH_2O$ for a final volume of 80 μL, and then mixed with 100 μL of 2X SEAP buffer (20 mM homoarginine (ThermoFisher catalog# H27387), 1 mM $MgCl_2$, and 21% (v/v) dioethanolamine (ThermoFisher, catalog# A13389)) and 20 μL of the p-nitrophenyl phosphate (PNPP, Acros Organics catalog# MFCD00066288) substrate (120 mM). Samples were measured via kinetic measurements (1 measurement/min) for a total of 30 minutes at 405 nm using a SpectraMax iD3 spectrophotometer (Molecular Devices) with the Softmax pro software (version 7.0.2).

Secreted GFP was measured by incubating cell-free supernatant with cells displaying the Gbp6 anti-GFP-binding nanobody, with mCherry fused at the C-terminus, which acted as a co-transfection marker (Supplementary Fig. 10a). The Gbp6 anti-GFP nanobody expressing cells were incubated for 2 h with supernatant from various RELEASE conditions and then analyzed using flow cytometry. To quantify changes in the amount of GFP secretion, we selected and compared the median GFP fluorescence from nanobody-displaying cells with the highest expression of the mCherry co-transfection marker. Supernatant from cells transfected that constitutively secrete SEAP, or GFP were used as negative and positive controls, respectively (Supplementary Fig. 10b).

To measure the amount of secreted IL-12, cell-free supernatant was collected and quantified using the Human IL-12p70 DuoSet ELISA (R&D Systems; catalog# DY1270), as per the manufacturer's instructions.

**Flow cytometry and data analysis**. Two days after transient transfection, cells were harvested using FACS buffer (HBSS + 2.5 mg/mL of Bovine Serum Albumin (BSA)). For experiments requiring antibody staining, surface GFP was measured by incubating cells with a 1:1000 dilution of anti-GFP Dylight 405 antibody (ThermoFischer; catalog# 600-146-215) in FACS buffer for one hour at 4 °C. For experiments measuring the surface display of Kir2.1, cells were incubated with 1:500 dilution of anti-hemagglutinin antibody (HA, Abcam; catalog# ab137838), followed by incubation with 1:1000 dilution of a donkey anti-rabbit IgG conjugated to alexa-647 (Abcam, catalog# ab150075). After staining, cells were washed twice with FACS buffer and then strained using a 40 μm cell strainer. Cells were analyzed by flow cytometry (BioRad ZE5 Cell Analyzer) and Everest software (version 3.1). As previously described[7], we use the EasyFlow MATLAB-based software package developed by Yaron Antebi (https://github.com/AntebiLab/easyflow.git) to process the flow cytometry data.

For analysis, we selected and compared cells with the highest expression of the co-transfection marker, which was typically mCherry (Supplementary Fig. 5). This was done to have the largest separation between basal reporter autofluorescence from cellular autofluorescence, as previously described[7,26]. For experiments using the Kir2.1 potassium channel, cells were either co-transfected with the voltage indicator ASAP3[57] or incubated with the Oxonol chemical dye, DiSBAC$_2$(3) (20 μM in HBSS) for 5 min before performing flow cytometry[58]. The N-terminus of Kir2.1 was fused with mCherry, or GFP, which acted as a co-transfection marker. After gating on cells with high expression of Kir2.1, the median fluorescence intensity was used to estimate changes in membrane potential[58].

**Statistical analysis**. Values are reported as the means from at least 3 biological replicates, which was representative from two independent biological experiments. For experiments comparing two groups, an unpaired Student's $t$-test was used to assess significance, following confirmation that equal variance could be assumed ($F$-test). If equal variance could not be assumed, then a Welch's correction was used. For experiments comparing three or more groups, a one-way ANOVA, or a two-way ANOVA with a post hoc Tukey test was used to compare the means among the different experimental groups. Data were considered statistically significant at a $p$-value of 0.05. Data are presented as average ±SEM, unless otherwise stated. All statistical analysis was performed using Prism 7.0 (GraphPad).

**Reporting summary**. Further information on research design is available in the Nature Research Reporting Summary linked to this article.

## Data availability

New plasmids used in this study will be made available for distribution from Addgene (https://www.addgene.org/Xiaojing_Gao/). Annotated plasmid sequences used in this study are provided in the Source Data as GeneBank files. Raw.fcs files are available from the corresponding authors upon reasonable request. Raw experimental data and p-values for each figure are provided as Source Data. Source data are provided with this paper.

## Code availability

EasyFlow MATLAB code used for flow cytometry analysis is available from the GitHub repository at https://github.com/AntebiLab/easyflow.git.

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

## Acknowledgements
We would like to thank Dr. Lin, and Dr. Leonard for kindly sharing some of their plasmids that were used in this work. We would also like to thank Leo Scheller for providing the protocol for the SEAP assays. This work was funded by NIH (4R00EB027723-02, X.J.G), Stanford Cancer Institute (Cancer Innovation Award #216174, X.J.G.), Stanford SystemX Alliance (Seed Grant, X.J.G.), NSERC (PDF-557516-2021, A.E.V.), the International Human Frontier Science Program Organization (LT000221/2021-L, A.E.V.), and the Stanford Graduate Fellowship (J.K. and C.A.).

## Author contributions
A.E.V. and X.J.G. conceived and directed the study. A.E.V, J.K, and C.A. performed most of the experiments. X.J.G. created the HRAS-sensing protease, and L.S.C and R.Z. created the protease-activated protease under M.B.E.'s supervision. A.E.V, and J.K. analyzed the data for the manuscript. A.E.V., J.K. and X.J.G. wrote the manuscript. All authors provided feedback on the manuscript.

## Competing interests
The board of trustees of the Leland Stanford Junior University have filed a patent on behalf of the inventors (A.E.V., J.K., and X.J.G.) of the RELEASE platform described (US provisional Application No. 63/282689). The remaining authors declare no competing interests.
