## [Peer Review File · Nature Communications]

Reviewers' Comments:

Reviewer #1:

Remarks to the Author:

Comments:

The authors developed a system, Retained Endoplasmic Cleavable Secretion (RELEASE), allowing engineered proteins to be kept in the endoplasmic reticulum and only displayed/secreted in response to specific proteases. By linking RELEASE to other sensing circuits, the authors can control protein secretion in response to the "undruggable" oncogene KRAS mutants. As such, RELEASE should allow a local and programmable delivery of intercellular cues. This is a novel technology with potentially powerful applications in the future. However, some concerns may need to be addressed for your consideration.

Main:

1. Compared to traditional inducible gene expression system such as doxycycline inducible secretory protein expression, what's the advantage of the RELEASE system? This needs to be clearly established, ideally by head-to-head comparing the two systems.
2. It will be informative if the authors can show how fast the protein can be secreted or displayed on the membrane upon cleavage. Maybe microscope images showing the cleavage and transporting process would be helpful to understand this dynamic process.
3. Using RELEASE circuits to sense cancerous cells (Ras signal in this paper) and secrete inflammation cytokines (IL-12 in this paper) has been demonstrated in the manuscript in HEK cells. Could the authors explain more on how to apply this for real cancer diagnosis/therapeutics situations? For example, how to engineer the cancer cells in patients? Whether the induced production of IL-12 will be sufficient in copy numbers for therapeutic purpose? Experiments will be needed to demonstrate and establish this concept with convincing results.

Minor:

- a. Some of the cartoons are confusing and hard to understand. For example, color codes in the legend of figure 1a. Also, please explicitly indicate what are the gray circles, red circles and orange circles in figure 4a.
- b. Add statistical analysis to supplementary figures 4-7.

Reviewer #2:

Remarks to the Author:

This manuscript "Protease-controlled secretion and display of intercellular signals" by Vlahos and colleagues reports on the development of a methodology, named RELEASE, to control the secretion or surface display of proteins expressed into the secretory pathway in mammalian cell lines using orthogonal proteases such as TEVp. The manuscript is clearly written and easy to understand. The experimental results are well-presented, concise and follow a logical order. The paper demonstrates a novel protein-based module that could be used to enhance the control of protein secretion and display that may have potential in a range of biomedical applications. The authors demonstrate their potential to secrete/display a range of proteins and combine modules to form logic gates and signal relays. The conclusions of the paper are valid based on the presented results, though I have a few comments regarding the need to include DNA sequences and constructs and the need to include more methodological details for reproducibility, as well as regarding missing experiments to validate that increased secretion in the absence/presence of proteases or other elements is not due to uncontrolled variables, such as differences in protein levels 'before' secretion.

Comments on data, experiments, and discussion

1. Line 137 and throughout the paper. The authors mention the proteins they used but say next to nothing about the structure or sequences of their constructs, which is odd for a synthetic biology paper about circuit based control. While protein based circuits may make the circuit less reliant on certain transcriptional aspects, the sequences of the DNA constructs ought to be provided in full, and the plasmid maps also ought to be provided in the Supplementary Figures. This is required for openness and reproducibility but also for readers to understand how these circuits might work – are the transcriptional units constitutive or inducible? How many parts are required for a functional

circuit? etc. In the methods section there is a single comment about inducer usage but in Supplementary Table 2 it seems some experiments used multiple plasmids, suggesting there's missing information.

2. Fig 1c,d and throughout the paper. The authors have chosen to make their figures clean of details which is fine, but the legends ought to include missing salient experimental details, e.g. what is meant by protease "+"? Which inducer was used and how much was used? How long after induction was the assay done, and which assay was used?

3. Line 139 and throughout the paper. How was SEAP measured? While the details are in the Methods, this should be clearly stated at least the first time the authors refer to the results of an assay.

4. Line 139 and throughout the paper. Assays to control for protein levels are missing from the paper. How do the authors know that discrepancies in protein levels haven't caused the phenotypes they are seeing? Synthetic circuit parts can sometimes affect one another, so protein level controls are important. This may be unlikely for a small mutation but possible for a condition using +/- protease expression. If protein levels increased overall, secretion would also be expected to increase. The authors should justify in the text why this wasn't included, or include this data as control figures.

5. Line 223 and Supplementary Fig. 5 and Fig 3d. Again, as this is the first time that flow cytometry is used as an assay, the authors should describe briefly in the text and legend how the assay was performed. They should also present the flow data in the supplementary, along with gating decisions and a brief explanation of how it was analysed. Flow data is usually quite noisy, so visualising distributions is important even if the median is the only value used in the main figures.

6. Line 239. The authors need to explain what they mean by "The chemical dye DiSBAC2(3) showed similar results (Fig. 3f), and the observed change in median fluorescent intensity was indicative of a 30 mV change in membrane potential" as it is unclear how fluorescence intensity measured in arbitrary units could allow them to suggest a change in mV units. I assume this assay involves adding DiSBAC2 to cells, washing off excess and using flow cytometry to quantify the level of dye per cell, but again this ought to be mentioned in the results so readers understand. They should also explain how changes in ASAP3 and DiSBAC2 uptake could prove hyperpolarisation in the absence of positive control conditions which are already known to hyperpolarise cells and are missing from the figures.

7. Line 259. The use of "HRAS-G12V" is made without introduction or explanation as to what this mutant is/does. Is it constitutively activated? Relatedly, there is also no explanation in the text or legend of Supplementary Fig. 6 of what is meant by "g0". Noticing later that "g0" seems to refer to endogenous KRAS, i.e. wt cells, my interpretation of the results in Supplementary Fig. 6 is not that TEVP causes a minimal increase, but either that endogenous KRAS is activated to a similar level to the mutant, or that the split TEVP has a high level of self-activation in the absence of active KRAS.

8. Line 420. The authors have a highly unusual way of quantifying secreted GFP, seemingly via binding to a cell displaying an anti-GFP nanobody. If this method has not been described elsewhere, the authors should add the plasmid used in this assay to the list of plasmids used, and describe in detail how GFP binding to these cells is quantified – is it done with flow cytometry? How is the data analysed and normalised? Again, raw flow data (histograms/violin plots) should be supplied in the supplementary figures.

9. Discussion. The authors have shown that RELEASE can be used as a module in mammalian synthetic circuits, however, these results are not the equivalent of showing they will work in vivo. The authors do not discuss for example the likely challenges of moving these systems in vivo, the most basic of which would require a demonstration that endogenous levels of (e.g.) KRAS can induce significant differences between healthy cells and cells with elevated levels of activated KRAS. The authors should mention this as a required intermediate step to move towards application.

Minor comments

10. Line 125. The first sentence will not be accessible to readers unfamiliar with the mammalian secretory pathway jargon. It may be helpful to replace 'luminal' with 'ER-facing' or 'ER luminal', and to preface 'ER retention motif' with 'cytoplasmic ER retention motif'.

11. Line 132. As above, perhaps replace "After the first cytosolic cleavage event, the membrane-tethered protein is processed into its soluble form" with "After the first cytosolic cleavage event and transport into the trans-Golgi, the membrane-tethered protein is processed into its soluble

form.”

12. Fig 1a. The key is useful but should be coloured, and to avoid confusion, explicitly labelled, i.e. “TEV protease” not just “protease”, “TEV protease cut site” not just “protease cut site” – since furin is also a protease.

13. Fig 1c. Is the bottom diagram correct? It suggests the not retained version is not secreted.

14. Line 140. How was the dilysine motif mutated?

15. Fig. 2 and all similar figures. Again, the authors shouldn’t assume that readers have memorised what their depictions mean. Either a legend that connected shapes/colours to their protein domains, or direct labels, should be used for each figure. For example, presumably the salient parts of Fig 2c are the two protease sites. These could at least be labelled with what they represent, and the same for Fig. 2d protease sites and retention sequences. Labels as simple as “TEVp” or “TVMVp” next to the orange/yellow/blue pacman images would also help readers understand the figures.

16. Line 218. “Unlike secreted proteins, Kir2.1 has cytosolic motifs that directs its transport in the secretory pathway, posing unique challenges for RELEASE and serving as a test case for its future adaptation to other membrane proteins.” It is unclear what is meant by Kir2.1 posing unique challenges.

17. Line 218 and Supplementary Figure 5. The topology of Kir2.1 is unclear. The authors might consider adding a diagram akin to that in Fig1b for Kir2.1, to illustrate where the ER retention signal was placed, and what they mean by the C terminus being distal to the ER membrane.

18. Line 234. The authors might consider clarifying the phrase “DiSBAC2(3) is a chemical dye that decrease cell entry”. I assume they mean its diffusion into cells is inhibited by hyperpolarisation?

19. Figure 5. Missing “circuit” in “Each component of the engineered can ..”.

We thank the reviewers for their excellent and insightful comments. We believe we have addressed all the comments and strengthened the quality of the manuscript. Point-by-point responses are given below, and the manuscript has been updated accordingly.

Reviewer #1 (Remarks to the Author):

Comments:

The authors developed a system, Retained Endoplasmic Cleavable Secretion (RELEASE), allowing engineered proteins to be kept in the endoplasmic reticulum and only displayed/secreted in response to specific proteases. By linking RELEASE to other sensing circuits, the authors can control protein secretion in response to the “undruggable” oncogene KRAS mutants. As such, RELEASE should allow a local and programmable delivery of intercellular cues. This is a novel technology with potentially powerful applications in the future.

Response: We thank the reviewer for this comment and agree that this new technology may lead to many exciting new applications.

Main:

1. Compared to traditional inducible gene expression system such as doxycycline inducible secretory protein expression, what’s the advantage of the RELEASE system? This needs to be clearly established, ideally by head-to-head comparing the two systems.

Response: We thank the reviewer for this comment and agree with the importance of explaining the advantages of the RELEASE system and comparing it to traditional inducible gene expression systems. Traditional inducible transcriptional systems, such as those induced by doxycycline, allow users to control gene expression using externally administered small molecules, but cannot express genes in response to intracellular/extracellular states. In contrast, RELEASE can be activated in response to any multimerization event, not only small molecule-induced dimerization, but also signaling-dependent protein dimerization that represent cellular states. This allows RELEASE to create a sense-and-respond closed circuit within a cell, without requiring exogenous activators such as doxycycline to function. Therefore, RELEASE sets the foundation for our vision of “circuits as medicine”, the Ras-to-cytokine circuit in the paper being an example, which wouldn’t be possible with any small-molecule induced transcriptional reporter. We have added a sentence at in the first paragraph of discussion to articulate that (lines 361-364).

Furthermore, even if just inducing RELEASE with a small molecule, because RELEASE operates at the protein level in contrast to the transcriptional control of dox-based systems, they can synergize to achieve more stringent control. No inducible system is 100% leak-proof, and controlling at two orthogonal levels would further reduce such leak. This could be important for potent intercellular signals, such as surface T-cell engagers that must be tightly regulated. We thank the reviewer for prompting us to explore this feature, and we have now validated that combining RELEASE and dox achieves a lower leaky baseline than either alone. We have added the dual-induction experiment to figure 1 (**Fig. 1f**) and have modified the text accordingly:

“Dual-input systems for fine tuning signaling pathways are an important tool in synthetic biology¹ and we hypothesized that RELEASE could be used with traditional inducible gene expression systems to achieve this. Since RELEASE controls protein secretion post-translationally, we used a doxycycline-inducible (DOX) SEAP-RELEASE construct, co-expressed with a rapalog-inducible split-TEVP and measured SEAP secretion with different combinations of inducers (**Fig. 1f**). Indeed, SEAP secretion was dependent on the presence of both inducers, and the biggest increase in SEAP secretion was observed when both rapalog and DOX was present relative to when they were both absent (**Fig. 1f** – 76.9-fold increase).” (lines 162 – 169).

“RELEASE is also compatible with traditional gene expression systems (i.e. DOX) to form a dual-input system to offer stricter dynamic control over protein secretion or surface expression (**Fig. 1f**).” (lines 369-371)

Finally, as the reviewer pointed out, post-translational and transcriptional controls might exhibit distinct temporal dynamics. We also compared the dynamics of protein secretion using a DOX inducible traditional gene expression system and RELEASE in response to the following question.

2. It will be informative if the authors can show how fast the protein can be secreted or displayed on the membrane upon cleavage. Maybe microscope images showing the cleavage and transporting process would be helpful to understand this dynamic process.

Response: We thank the reviewer for this comment and agree that showing the dynamics of RELEASE is important. To determine the speed of protein secretion using RELEASE, we co-transfected cells with SEAP-RELEASE and a rapalog-inducible split-TEVP. 48 hours after transfection we induced the cells with rapalog and measured SEAP secretion over 6 hours. Using RELEASE, we observed a significant difference in protein secretion beginning at 3 hours post induction. To compare RELEASE with traditional inducible gene expression systems, we transfected cells with SEAP alone under the control of a DOX-inducible CMV promoter (CMV-TO). Like RELEASE, we observed a significant difference in protein secretion beginning at 3 hours post induction with DOX.

Intuitively, one might expect traditional inducible gene expression systems to be slower than RELEASE, because the former must undergo transcription and translation to secrete proteins, compared to the latter that have already been translated and rely on post-translational modifications. Several factors and hypotheses might explain this counterintuitive result. First of all, it has been established that it takes ~30-60 minutes for proteins to transition from ER to trans Golgi², and further transport to the cell membrane would take longer; it also takes time for rapalog to diffuse into the cells. All these “temporal overheads” set an upper limit on the speed of RELEASE, so we would not expect to observe any difference within one hour of inducer addition. Second, several steps of RELEASE rely on endogenous proteins, which might be saturated and pose rate-limiting bottlenecks. For example, unlike the “plain” SEAP regulated by dox, SEAP-RELEASE must undergo an additional proteolytic cleavage event by furin in the Golgi Apparatus. We hypothesized that we could improve the speed of protein secretion by overexpressing the furin endoprotease. Indeed, cells overexpressing furin showed a significant increase in SEAP secretion beginning at 2 hours post induction, already approaching the aforementioned speed limit. This suggested that we could improve the speed of protein secretion by improving the cleavage efficiency of the furin cut site, such as using different furin cut sites or placing multiple furin cut sites in tandem.

We have modified the results and discussion to reflect this:

“We also compared the dynamics of SEAP secretion using RELEASE to traditional inducible gene expression systems (**Supplementary Fig. 4a**). Following transfection using a DOX-inducible SEAP, there was a significant difference in protein secretion observed 3 hours after stimulation with DOX, compared to the uninduced controls (**Supplementary Fig. 4b**). In comparison, SEAP-RELEASE co-expressed with a rapalog-inducible split-TEVP showed a significant difference in protein secretion, 3 hours post-induction with rapalog (**Supplementary Fig. 4c**), which could be improved to 2 hours by overexpressing furin (**Supplementary Fig. 4d**). This suggested that furin cleavage could be one of the factors limiting the speed of protein secretion using RELEASE.” (lines 170-178).

“If the new application requires a faster response, we could improve the cleavage efficiency of the furin cut site of RELEASE, such as using different furin cut sites or using a protein linker with multiple furin cut sites in tandem, as furin was implicated to be a limiting factor in the speed of RELEASE (Supplementary Fig. 4c, 4d)” (lines 376-379).

Supplementary Figure 4: Dynamics of protein secretion using traditional inducible gene expression systems and RELEASE.

a) Schematic of cells transfected with DOX-inducible SEAP, and SEAP-RELEASE and rapalog-inducible split-TEVP. 48 hours after transfection, cells were induced with DOX or rapalog, and the supernatant was collected. **b)** A significant difference in SEAP secretion was first observed 3 hours post-induction between cells induced with DOX and the uninduced controls. **c)** Using SEAP-RELEASE, a significant difference between induced and uninduced cells was observed beginning at 3 hours post-induction with rapalog. **d)** Through overexpression of the furin endoprotease, increased SEAP secretion was observed as quickly as 2 hours after induction using Rapalog compared to cells induced with the vehicle. The vehicle for rapalog was 95% EtOH. Each dot represents a biological replicate. Mean values were calculated from four biological replicates (**b, c, d**) +/- SEM. The results are representative of at least two independent experiments; significance was tested using by two-way ANOVA with a Tukey's post-hoc comparison test among the multiple conditions. *** = $p < 0.001$, **** = $p < 0.0001$.

3. Using RELEASE circuits to sense cancerous cells (Ras signal in this paper) and secret inflammation cytokines (IL-12 in this paper) has been demonstrated in the manuscript in HEK cells. Could the authors explain more on how to apply this for real cancer diagnosis/therapeutics situations? For example, how to engineer the cancer cells in patients? Whether the induced production of IL-12 will be sufficient in copy

numbers for therapeutic purpose? Experiments will be needed to demonstrate and establish this concept with convincing results.

Response: We agree with the reviewer that it is an exciting new direction that we are exploring, and a substantial amount of work will be expected. We thank the reviewer for offering us the opportunity to elaborate on our vision. One of the applications that we are currently pursuing for RELEASE is cancer immunotherapy. To apply this for real cancer diagnosis/therapeutics we anticipate two problems that must be overcome: the first being delivery and the second being selectivity. One of the advantages of using protein-based circuits and RELEASE is that they can be encoded within single mRNA transcript³, using multiple self-cleaving peptides (i.e. P2A, T2A)⁴, or internal ribosomal entry sites (IRES) to separate components. Another parameter for delivering our therapeutic is delivery efficiency. We anticipate that current delivery approaches will not be suitable for delivering the therapeutic circuit into every cancer cell and have chosen to pursue using RELEASE as a tool for cancer immunotherapy instead of direct cancer cell killing. We hypothesize that our therapeutic circuits will be able to emulate the abscopal effect⁵, where tumor burden can be reduced through stimulating and training the host's immune system without having to transfect every cancer cell – however this is currently outside the scope of this manuscript.

The other problem that must be overcome is specificity, ensuring that the therapeutic circuit only activates in cancer cells and not healthy cells. This is critical since many proteins and pathways affected in cancerous cells are normally present in healthy cells as well. Selecting proteins that are either overexpressed or constitutively activated that could be targeted through protein-binding events will be required to develop suitable protease-based sensors. For example, the RAS pathway is tightly regulated in normal cells, and we opted to use a RAS sensor (RBD-binding) dependent on RAS activation. As a first step, we observed that our RBD-split TEVP sensor could increase protein secretion in cells expressing exogenous KRAS active mutants (**Fig. 4e**), relative to exogenous wildtype KRAS, which would be subject to endogenous regulation. Many of the RAS+ cancers also have additional mutations to tumor suppressor genes, such as p53, which may allow us to use AND logic with RELEASE to improve specificity.

Showing the feasibility of this sensor using HEK293 cells expressing KRAS active mutants was the first step, and the next step will be to validate the sensor using various RAS mutant cancer cell lines (i.e. LS180, PANC 10.05, or PANC 03.27). With additional proteases we can effectively threshold the activation of our therapeutic circuit ensuring that it will not activate in normal healthy cells. The levels of each protein circuit component will have to be tuned using cancer cell lines. After validating our RAS-sensing circuit using cancer cell lines, the next step will be to validate its efficacy in vivo using a xenograft subcutaneous tumor model, however study is currently ongoing and outside the scope of this work.

Using IL-12 fused to RELEASE, we were able to get a comparable concentration of secreted IL-12 with previously reported work⁶ that showed to be therapeutically relevant using an in vivo tumor model. We have added a sentence in results to reflect this (lines 236-238).

We have added the following text in the discussion to outline the workflow we anticipate will need to be done to apply RELEASE and protein circuits for cancer immunotherapy:

“The selectivity of the circuit will be further improved using additional proteases through quantitative thresholding (**Fig. 5g**) or logic operations. For the latter, many RAS-driven cancers harbor additional mutations to tumor suppressor proteins, such as p53. One could use split proteases fused to nanobodies that have preferential binding to mutant p53, to activate RELEASE only when both mutant KRAS and mutant p53 are simultaneously present, via AND logic (**Fig. 2c**). In this work, we used HEK293 cells expressing exogenous wildtype or active mutant KRAS to demonstrate that our therapeutic circuit could

distinguish between them (**Fig. 4e**). However, future studies must be completed using cancer cells harboring the active KRAS mutants (i.e. PANC 10. 05, PANC 03. 27, or LS180) to demonstrate significant differences with endogenous KRAS activity in healthy cells, in vitro and using in vivo xenograft models.” (lines 383 – 393)

“Delivery will be another barrier that will have to be overcome for RELEASE to be useable for cancer immunotherapy. An additional benefit compared to traditional synthetic circuits, is that protease circuit components can be encoded within single mRNA transcripts³ that do not pose the risk of insertional mutagenesis. Furthermore, combinatorial immunomodulators have been found to be more effective than monotherapies for targeting cancer cells. Due to complexity and broad activity of the immune system, we anticipate that we will need to control the secretion and surface display of multiple immunostimulatory proteins, which will be possible due to the modularity of RELEASE (**Fig. 2b, 3b**).” (lines 394 – 401).

Minor:

a. Some of the cartoons are confusing and hard to understand. For example, color codes in the legend of figure 1a. Also, please explicitly indicate what are the gray circles, red circles and orange circles in figure 4a.

Response: We have updated the color codes in the legend of figure 1a, and included a legend for figure 4a.

b. Add statistical analysis to supplementary figures 4-7.

Response: Statistical analysis has been added to supplementary figures 4-7.

Reviewer #2 (Remarks to the Author):

This manuscript “Protease-controlled secretion and display of intercellular signals” by Vlahos and colleagues reports on the development of a methodology, named RELEASE, to control the secretion or surface display of proteins expressed into the secretory pathway in mammalian cell lines using orthogonal proteases such as TEVp. The manuscript is clearly written and easy to understand. The experimental results are well-presented, concise and follow a logical order. The paper demonstrates a novel protein-based module that could be used to enhance the control of protein secretion and display that may have potential in a range of biomedical applications. The authors demonstrate their potential to secrete/display a range of proteins and combine modules to form logic gates and signal relays. The conclusions of the paper are valid based on the presented results, though I have a few comments regarding the need to include DNA sequences and constructs and the need to include more methodological details for reproducibility, as well as regarding missing experiments to validate that increased secretion in the absence/presence of proteases or other elements is not due to uncontrolled variables, such as differences in protein levels ‘before’ secretion.

Response: We thank the reviewer for this comment and agree that RELEASE is a novel protein-based modules that has potential for a range of biomedical applications. Please find point-by-point responses below for your comments. We believe that your comments have helped improve the clarity for the methodology used in this work and highlights the benefits of RELEASE.

Comments on data, experiments, and discussion

1. Line 137 and throughout the paper. The authors mention the proteins they used but say next to nothing about the structure or sequences of their constructs, which is odd for a synthetic biology paper about circuit based control. While protein based circuits may make the circuit less reliant on certain transcriptional aspects, the sequences of the DNA constructs ought to be provided in full, and the plasmid maps also ought to be provided in the Supplementary Figures. This is required for openness and reproducibility but also for readers to understand how these circuits might work – are the transcriptional units constitutive or inducible? How many parts are required for a functional circuit? etc. In the methods section there is a single comment about inducer usage but in Supplementary Table 2 it seems some experiments used multiple plasmids, suggesting there's missing information.

Response: We thank the reviewer for this comment and agree that one of the goals of our manuscript is to make RELEASE accessible to the general scientific community using protein-based circuits. All the plasmids used in this study are under the control of the CMV-TO promoter, which requires doxycycline for transcriptional activation. In our study, doxycycline was administered at the time of transfection so that constructs were effectively “constitutively expressed”. Since all the constructs used in our study are under the control of the CMV-TO promoter, we do not anticipate any differences in transcriptional activation among the different plasmids.

To control protein secretion using RELEASE, all one needs is the plasmid containing the RELEASE construct and another plasmid containing the cognate protease (or sensor activating protease). Additional components can be implemented depending on the specific application. The reviewer is correct that some experiments used multiple plasmids, such as those coding for different RELEASE variants, different proteases, or sensors and we have outlined them in supplementary table 2.

We are depositing all new plasmids and their associated DNA sequences used in our study on Addgene, which will be released upon publication, and we added all the DNA sequences for our plasmids into the supplementary text.

“A complete list of plasmids used for each experiment (**supplementary table 2**) and the respective amounts used for transfections can be found in **supplementary table 3**. In addition, the DNA sequences of all the plasmids used in this study can be found in the source data, and all new plasmids and sequences are available on Addgene.” (lines 435 – 438).

2. Fig 1c,d and throughout the paper. The authors have chosen to make their figures clean of details which is fine, but the legends ought to include missing salient experimental details, e.g. what is meant by protease “+”? Which inducer was used and how much was used? How long after induction was the assay done, and which assay was used?

Response: We thank the reviewer for this comment and apologize for not being clear in the manuscript. We have added a new supplementary table (supplementary table 2), which outlines all the salient experimental details for each experiment. When we include or drop a protein component in our circuit, we do so by including or dropping the plasmid that encodes it in our transfection.

“A complete list of plasmids used in this study (**supplementary table 2**) and the respective amounts used for transfections can be found in **supplementary table 3**.” (lines 435-436)

3. Line 139 and throughout the paper. How was SEAP measured? While the details are in the Methods, this should be clearly stated at least the first time the authors refer to the results of an assay.

Response: We thank the reviewers for their comment. Unless otherwise stated, SEAP activity was measured by taking the supernatant from cells 48 hours after transient transfection. We have modified line 139 to reflect this:

“We transiently transfected human embryonic kidney (HEK) 293 cells using DNA plasmids encoding the constructs and 48 hours later removed the supernatant to measure SEAP activity following incubation with a colorimetric substrate.” (lines 136 – 138).

“All measurements were taken 48 hours after transient transfection, unless otherwise stated.” (lines 448 – 449)

4. Line 139 and throughout the paper. Assays to control for protein levels are missing from the paper. How do the authors know that discrepancies in protein levels haven't caused the phenotypes they are seeing? Synthetic circuit parts can sometimes affect one another, so protein level controls are important. This may be unlikely for a small mutation but possible for a condition using +/- protease expression. If protein levels increased overall, secretion would also be expected to increase. The authors should justify in the text why this wasn't included or include this data as control figures.

Response: We thank the reviewer for this comment and agree that synthetic parts can sometimes affect one another. The reviewer is correct that by increasing the protein levels (i.e. by transfecting a greater amount of plasmid DNA), we would also observe a greater amount of secretion, however we have found this to only be the case when increasing the plasmid amount of the RELEASE construct when transfecting. In our experiments, we observed a slight reduction in protein secretion of both GFP and SEAP by increasing protein levels (co-expression of TEVP and HCVP), relative to the conditions where only a single protease was expressed (**Fig. 2a, b**). In addition, when overexpressing a protease that did not match the cleavage site in RELEASE, we did not detect any change of SEAP baseline (Fig. 2a, b). These results suggest that indeed there might be some resource competition, where overexpressing a protease would mildly **decrease** SEAP/GFP. Therefore, the protease-dependent SEAP/GFP **increase** is very unlikely to be a non-specific artifact. We have now highlighted this in the text:

“In addition, we observed a slight reduction in SEAP and GFP secretion when both proteases were expressed, relative to when only a single protease was co-expressed, potentially due protein overexpression draining finite cellular resources.” (lines 198 – 201).

5. Line 223 and Supplementary Fig. 5 and Fig 3d. Again, as this is the first time that flow cytometry is used as an assay, the authors should describe briefly in the text and legend how the assay was performed. They should also present the flow data in the supplementary, along with gating decisions and a brief explanation of how it was analysed. Flow data is usually quite noisy, so visualising distributions is important even if the median is the only value used in the main figures.

Response: We thank the reviewer for this comment and agree that we should have included a supplementary figure outlining our gating decisions for the flow cytometry data. We have also included distribution of the flow cytometry data from each experiment in the supplementary text.

“To validate this strategy, membrane-bound green fluorescent protein (GFP) fused to RELEASE and a mCherry co-transfection marker was transfected into HEK293 cells, and the cell surface was stained using an anti-GFP antibody. To quantify changes in the surface display of GFP, the mCherry co-transfection marker was used to select for highly transfected cells, which show the largest separation of reporter fluorescence from cellular autofluorescence (**Supplementary Fig. 5**).” (lines 181 – 186).

“To measure the surface display of Kir2.1, a hemagglutinin (HA) epitope was incorporated into its extracellular loop³⁴. In addition, mCherry, or GFP was fused to the N-terminus of Kir2.1, to select cells that were highly expressing Kir2.1 for flow cytometry analysis (**Supplementary Fig. 7a**).” (lines 247 - 250).

Supplementary Figure 5: Gating strategy for flow cytometry analysis. a) Cells were gated based on FSC-A and SSC-A, followed by gating for singlets **b)**. Each experiment uses a co-transfection marker, such as mCherry to select for highly transfected cells (~5% of the cellular population) for analysis. **d)** Representative experiment analyzing the amount of surface GFP expression using RELEASE with and without TEVP (data for **Fig. 1h**). The green and red stars represent the median fluorescence intensities of cells containing GFP fused to RELEASE with and without TEVP, respectively.

6. Line 239. The authors need to explain what they mean by “The chemical dye DiSBAC2(3) showed similar results (Fig. 3f), and the observed change in median fluorescent intensity was indicative of a 30 mV change in membrane potential” as it is unclear how fluorescence intensity measured in arbitrary units could allow them to suggest a change in mV units. I assume this assay involves adding DiSBAC2 to cells, washing off excess and using flow cytometry to quantify the level of dye per cell, but again this ought to be mentioned in the results so readers understand. They should also explain how changes in ASAP3 and DiSBAC2 uptake could prove hyperpolarisation in the absence of positive control conditions which are already known to hyperpolarise cells and are missing from the figures.

Response: We thank the reviewer for this comment and apologize for not being clear in the manuscript regarding the methodology of the chemical dye, DiSBAC2(3). The reviewer is correct that the assay involves incubating cells with DiSBAC2(3), washing off the excess and then quantifying the level of dye uptake per cell using flow cytometry. It has been previously established that the uptake of DiSBAC2(3) is

inhibited in hyperpolarized cells⁷, and we indeed observed a reduction in the median fluorescence intensity in cells expressing Kir2.1 RELEASE and TEVP, relative to cells without TEVP (**Fig. 3f**). We are confident that the Kir2.1 is indeed localized to the surface when co-expressed with the cognate protease (HA staining, Fig. 3d). In addition, both the previously reported genetic encoded voltage ASAP3⁸, and the DiSBAC₂(3) chemical dye showed consistent results even though they measure changes in membrane potential (using fluorescence intensity as a readout) through two different mechanisms. However, we do agree that we may have overstated our statement regarding the absolute change in membrane potential, which was based on the mV-fluorescence calibration curves in published works and may not directly translate to our context. We have modified the sentence accordingly:

“ASAP3 is a genetically encoded voltage indicator⁵⁷ that increases fluorescence as cells become hyperpolarized, while DiSBAC₂(3) is a chemical dye that decreases diffusion into hyperpolarized cells, and therefore fluorescence intensity⁵⁹. When Kir2.1 RELEASE was co-expressed with ASAP3, we observed a significant increase in fluorescence intensity in response to TEVP (**Fig. 3e**), suggesting Kir2.1 was functional. The chemical dye DiSBAC₂(3) showed similar results (**Fig. 3f**), and the observed change in median fluorescent intensity that cells were hyperpolarized⁷, further corroborating that RELEASE-regulated Kir2.1 maintains its functionality.” (lines 262 – 269).

7. Line 259. The use of “HRAS-G12V” is made without introduction or explanation as to what this mutant is/does. Is it constitutively activated? Relatedly, there is also no explanation in the text or legend of Supplementary Fig. 6 of what is meant by “g0”. Noticing later that “g0” seems to refer to endogenous KRAS, i.e. wt cells, my interpretation of the results in Supplementary Fig. 6 is not that TEVP causes a minimal increase, but either that endogenous KRAS is activated to a similar level to the mutant, or that the split TEVP has a high level of self-activation in the absence of active KRAS.

Response: We thank the reviewer for this comment and apologize for not being clear in our text about the g0 groups. To keep the amount of plasmid transiently transfected consistent among all groups, g0 is a plasmid that encodes a scrambled sequence that does not translate any functional protein. The reviewer’s interpretation that the g0 group refers to wildtype cells (i.e. endogenous RAS activity) is correct, however we believe that the lack of difference observed between the two groups is due to the lack of signal propagation from the cell membrane to the ER, and not due to endogenous RAS activity being comparable to cells transiently transfected with the mutant active RAS. We believe the latter is the case, since the use of an intermediate relay protease in figure 4c and 4e, resulted in a significant difference in SEAP secretion among wildtype cells, cells transiently transfected with KRAS (which is still subject to endogenous regulation within the cell) and the active mutant KRAS-G12V.

We have modified the figure and figure legends to provide more clarity among the different groups and have modified the text to include the following sentences to further clarify the active mutant of G12V:

“The activating RAS mutations (i.e. G12V, G12C, G12D...) have been implicated in ... “ (line 274).

“... initial experiments performed using the active mutant HRAS-G12V and the RBD-split TEVP... “ (lines 288-289).

“... and observed comparable SEAP secretion with wildtype cells (endogenous HRAS activity) when regulated by TEVP-responsive RELEASE (**Supplementary Fig. 8**). Since HRAS-G12V reconstitutes RBD-split TEVP at the cell membrane, and cleavage of RELEASE occurs at the ER, we hypothesized that additional protease components would be required to propagate the signal from the cell membrane to the ER (**Fig. 4a**).” (lines 289 – 294).

8. Line 420. The authors have a highly unusual way of quantifying secreted GFP, seemingly via binding to a cell displaying an anti-GFP nanobody. If this method has not been described elsewhere, the authors should add the plasmid used in this assay to the list of plasmids used, and describe in detail how GFP binding to these cells is quantified – is it done with flow cytometry? How is the data analysed and normalised? Again, raw flow data (histograms/violin plots) should be supplied in the supplementary figures.

Response: We thank the reviewer for this comment and apologize for not being clear in the manuscript and omitting the GFP-nanobody plasmid from our supplementary tables. The reviewer is correct that we quantified GFP secretion by incubating the supernatant of cells transfected with GFP-RELEASE with cells displaying an anti-GFP nanobody. The intracellular C-terminus of the anti-GFP nanobody is fused with mCherry so that we can select cells that have the greatest expression of anti-GFP nanobodies and then quantify the amount of GFP bound on their surface using flow cytometry. We have revised the materials and methods to describe the method of quantification and have included a supplementary figure containing the raw flow data.

“Secreted GFP was measured by incubating cell-free supernatant with cells displaying the Gbp6 anti-GFP-binding nanobody, with mCherry fused at the C-terminus, which acted as a co-transfection marker (**Supplementary Fig. 10a**). The Gbp6 anti-GFP nanobody expressing cells were incubated for 2 hours with supernatant from various RELEASE conditions and then analyzed using flow cytometry. To quantify changes in the amount of GFP secretion, we selected and compared the median GFP fluorescence from nanobody-displaying cells with the highest expression of the mCherry co-transfection marker. Supernatant from cells transfected that constitutively secrete SEAP, or GFP were used as negative and positive controls, respectively (**Supplementary Fig. 10b**).” (lines 472 – 480)

Supplementary Figure 10: **a)** Schematic of cells transfected with the Gbp6 anti-GFP nanobody conjugated to mCherry. **b)** Receiver cells were incubated with cell free supernatant from various RELEASE conditions to validate that they could capture GFP. Supernatants from cells that constitutively secreted SEAP or GFP alone, were used as negative and positive controls, respectively. **c)** Raw flow plots from **Fig. 2b**, quantifying the amount of captured GFP on the receiver cells that were incubated with supernatant from cells co-expressed with SEAP-RELEASE, GFP-RELEASE, and different combinations of the cognate proteases. The stars represent the median fluorescence intensity for each group.

9. Discussion. The authors have shown that RELEASE can be used as a module in mammalian synthetic circuits, however, these results are not the equivalent of showing they will work in vivo. The authors do not discuss for example the likely challenges of moving these systems in vivo, the most basic of which would require a demonstration that endogenous levels of (e.g.) KRAS can induce significant differences between healthy cells and cells with elevated levels of activated KRAS. The authors should mention this as a required intermediate step to move towards application.

Response: We thank the reviewer for this comment and agree that we could have outlined the next immediate step to move towards using RELEASE for cancer immunotherapy. Please refer to response to the third comment by Reviewer 1.

Minor comments

10. Line 125. The first sentence will not be accessible to readers unfamiliar with the mammalian secretory pathway jargon. It may be helpful to replace 'luminal' with 'ER-facing' or 'ER luminal', and to preface 'ER retention motif' with 'cytoplasmic ER retention motif'.

Response: We thank the reviewer for these comments and have made the suggested changes to improve the clarity.

11. Line 132. As above, perhaps replace "After the first cytosolic cleavage event, the membrane-tethered protein is processed into its soluble form" with "After the first cytosolic cleavage event and transport into the trans-Golgi, the membrane-tethered protein is processed into its soluble form."

Response: We have made the suggested changes.

12. Fig 1a. The key is useful but should be coloured, and to avoid confusion, explicitly labelled, i.e. "TEV protease" not just "protease", "TEV protease cut site" not just "protease cut site" – since furin is also a protease.

Response: We thank the reviewer for this comment. We have added colours to the legend and included more descriptive labels.

13. Fig 1c. Is the bottom diagram correct? It suggests the not retained version is not secreted.

Response: We thank the reviewer for this comment. Yes, the reviewer is correct that the bottom diagram of figure 1.c represents a RELEASE construct containing the c-terminal -AAXX sequence, which would not be retained and subsequently secreted. To improve the clarity of the figure, we have updated figure 1c to include labels for each diagram, which describes the original ER-retaining dilysine motif construct (KKXX motif) and the non-retaining version (AAXX motif). We have updated the text to be more clear:

"... and used a dilysine-lacking motif (-AAXX) as the negative control (not retained)." (lines 135 – 136).

14. Line 140. How was the dilysine motif mutated?

Response: We apologize for not being clear in our manuscript. The dilysine motif was not mutated and an additional construct was cloned switching the ER-retaining dilysine motif (KKXX) with the AAXX motif that should not confer ER retention. We have updated the text to improve the clarity:

"Switching the dilysine retention motif (-KKXX) with -AAXX significantly increases SEAP secretion (Fig. 1c). (lines 140 – 141).

15. Fig. 2 and all similar figures. Again, the authors shouldn't assume that readers have memorised what their depictions mean. Either a legend that connected shapes/colours to their protein domains, or direct labels, should be used for each figure. For example, presumably the salient parts of Fig 2c are the two protease sites. These could at least be labelled with what they represent, and the same for Fig. 2d protease sites and retention sequences. Labels as simple as "TEVp" or "TVMVp" next to the orange/yellow/blue pacman images would also help readers understand the figures.

Response: We thank the reviewer for this comment and have updated the figures to include labels for the different proteases and protease cut sites throughout the manuscript.

16. Line 218. “Unlike secreted proteins, Kir2.1 has cytosolic motifs that directs its transport in the secretory pathway, posing unique challenges for RELEASE and serving as a test case for its future adaptation to other membrane proteins.” It is unclear what is meant by Kir2.1 posing unique challenges.

Response: We thank the reviewer for this comment and apologize for not communicating clearly in the manuscript. Kir2.1 has a cytosolic ER export motif (**FCYENEV**), which is necessary and sufficient for proper trafficking to the plasma membrane⁹. We were initially concerned that the retention strategy outlined using RELEASE, would not be able to overcome the effect of the ER export motif, but adoption of the RXR motif was able to achieve sufficient retention.

We have modified the text in the manuscript to clearly outline what was meant by Kir2.1 posing unique challenges:

“Unlike secreted proteins, Kir2.1 has a cytoplasmic-facing ER export motif (FCYENEV)⁹ that directs its transport through the secretory pathway, posing a potentially unique challenge in the retention ability of RELEASE and will serve as a test case for its future adaptation to other membrane proteins.” (lines 244 – 247).

17. Line 218 and Supplementary Figure 5. The topology of Kir2.1 is unclear. The authors might consider adding a diagram akin to that in Fig1b for Kir2.1, to illustrate where the ER retention signal was placed, and what they mean by the C terminus being distal to the ER membrane.

Response: We thank the reviewer for this comment and have added an illustration of the Kir2.1 construct in **Supplementary Figure 7**. We have also modified the text to describe where the RXR motif was placed:

“Indeed, a RELEASE construct using the RXR motif at the C-terminus, improved retention (**Supplementary Fig. 7**), and successfully controlled its surface display using TEVP (**Fig. 3d**).” (lines 255 – 257).

“Supplementary Figure 7: Surface display of Kir2.1 was dependent on the ER retention motif used in the RELEASE construct. **a)** Schematic of Kir2.1 RELEASE. The hemagglutinin (HA) epitope was incorporated into the extracellular loop to measure the surface expression of Kir2.1 using flow cytometry. **b)** Due to the large cytoplasmic tail of Kir2.1, the C-terminus was farther away from the ER membrane relative to other RELEASE constructs. **c)** The RXR motif retains proteins better than the KKXX motif when the C-terminal is distal to ER membrane. Each dot represents a biological replicate. Mean values were calculated from three biological replicates (**c**) +/- SEM. The results are representative of at least two independent experiments; significance was tested using an unpaired two-tailed Student’s *t*-test between the two indicated conditions for each experiment. * = $p < 0.05$ ”

18. Line 234. The authors might consider clarifying the phrase “DiSBAC₂(3) is a chemical dye that decrease cell entry”. I assume they mean its diffusion into cells is inhibited by hyperpolarisation?

Response: We thank the reviewer for their comment and we have modified the text:

“... DiSBAC₂(3) is a chemical dye that decreases diffusion into hyperpolarized cells, and therefore fluorescence intensity.” (lines 263 – 264)

19. Figure 5. Missing “circuit” in “Each component of the engineered can ..”.

Response: This has been fixed.

References:

1. Pedone, E. *et al.* A tunable dual-input system for on-demand dynamic gene expression regulation. *Nat. Commun.* **10**, 4481 (2019).
2. Boncompain, G. *et al.* Synchronization of secretory protein traffic in populations of cells. *Nat. Methods* **9**, 493–498 (2012).
3. Kowalski, P. S., Rudra, A., Miao, L. & Anderson, D. G. Delivering the Messenger: Advances in Technologies for Therapeutic mRNA Delivery. *Mol. Ther.* **27**, 710–728 (2019).
4. Szymczak, A. L. *et al.* Correction of multi-gene deficiency in vivo using a single “self-cleaving” 2A peptide-based retroviral vector. *Nat. Biotechnol.* **22**, 589–594 (2004).
5. Abuodeh, Y., Venkat, P. & Kim, S. Systematic review of case reports on the abscopal effect. *Curr. Probl. Cancer* **40**, 25–37 (2016).
6. Nissim, L. *et al.* Synthetic RNA-Based Immunomodulatory Gene Circuits for Cancer Immunotherapy. *Cell* **171**, 1138–1150.e15 (2017).
7. Shapiro, H. M. Estimation of membrane potential by flow cytometry. *Curr. Protoc. Cytom.* **Chapter 9**, Unit 9.6 (2004).
8. Villette, V. *et al.* Ultrafast Two-Photon Imaging of a High-Gain Voltage Indicator in Awake Behaving Mice. *Cell* **179**, 1590–1608.e23 (2019).
9. Stockklauser, C., Ludwig, J., Ruppertsberg, J. P. & Klöcker, N. A sequence motif responsible for ER export and surface expression of Kir2.0 inward rectifier K⁺ channels. *FEBS Lett.* **493**, 129–133 (2001).

Reviewers' Comments:

Reviewer #1:

Remarks to the Author:

The authors have satisfactorily addressed all the concerns and made appropriate modifications to the manuscript. There are some minor changes remaining to be made.

1. In supplementary figure 4a, the cartoon for the Dox inducible system seems to be confusing. The cartoon suggests that TetR is released after adding Dox, but how the transcription is initiated? Is the TetR blocking the transcription?
2. I agree with the authors points in the rebuttal to question 3. However, the authors varied the amount of each components transfected to achieve the optimal signal-to-background ratio in vitro, which may be difficult in vivo. Nevertheless, this direction is still very exciting and worth exploring.
3. There are some typos and punctuation errors in the manuscript (the list is not exhaustive):
 - a. There are some sentences missing period after citations. For example, in line 80, "...deliver therapeutic outputs accordingly^{1,2} This...", there should be a period after "1,2".
 - b. In line 383, "possibly" should be "possibility".

Reviewer #2:

Remarks to the Author:

The authors have addressed all my original points. Thank you for the careful and thorough responses. I have no more comments and am happy to recommend publication!

We thank the reviewers for their excellent and insightful comments. We believe we have addressed all the comments and strengthened the quality of the manuscript. Point-by-point responses are given below, and the manuscript has been updated accordingly.

Reviewer #1 (Remarks to the Author):

The authors have satisfactorily addressed all the concerns and made appropriate modifications to the manuscript. There are some minor changes remaining to be made.

1. In supplementary figure 4a, the cartoon for the Dox inducible system seems to be confusing. The cartoon suggests that TetR is released after adding Dox, but how the transcription is initiated? Is the TetR blocking the transcription?

Response: We apologize for the lack of clarity in our figure. The reviewer is correct that TetR is blocking transcription and the construct is under the control of the CMV promoter. We have revised the figure to include the CMV promoter to better reflect plasmid design.

2. I agree with the authors points in the rebuttal to question 3. However, the authors varied the amount of each components transfected to achieve the optimal signal-to-background ratio in vitro, which may be difficult in vivo. Nevertheless, this direction is still very exciting and worth exploring.

Response: We thank the reviewer for their comment. We agree that the direction is very exciting and worth exploring.

3. There are some typos and punctuation errors in the manuscript (the list is not exhaustive):
a. There are some sentences missing period after citations. For example, in line 80, "...deliver therapeutic outputs accordingly^{1,2} This...", there should be a period after "1,2".

Response: This and the other sentences within the manuscript have been fixed.

b. In line 383, "possibly" should be "possibility".

Response: This has been fixed.

Reviewer #2 (Remarks to the Author):

The authors have addressed all my original points. Thank you for the careful and thorough responses. I have no more comments and am happy to recommend publication!

Response: We thank the reviewer for their thoughtful and helpful comments. We believe they have greatly improved the clarity and reproducibility of the manuscript.